



# Sensitivity of Hydrologic and Geologic Parameters on Recharge Processes in a Highly-Heterogeneous, Semi-Confined Aquifer System

Stephen R. Maples[1], Laura Foglia[2], Graham E. Fogg[2], and Reed M. Maxwell[3]

[1]Hydrologic Sciences Graduate Group, University of California, Davis, One Shields Ave., Davis, CA 95616, USA
[2]Department of Land, Air, and Water Resources, University of California, Davis, One Shields Ave., Davis, CA 95616, USA
[3]Department of Geology and Geological Engineering, Colorado School of Mines, 1500 Illinois St, Golden, CO 80401, USA

**Correspondence:** Stephen R. Maples (srmap@ucdavis.edu)

**Abstract.**

Increasing reliance on groundwater resources has been observed worldwide during the past 50–70 years and has led to unsustainable groundwater abstraction in many regions, especially in semi-arid and arid alluvial groundwater basins. Managed aquifer recharge (MAR) has been promoted to replenish overdrafted groundwater basins and augment surface water sup-
ply. However, MAR feasibility in alluvial groundwater basins is complicated by complex geologic architecture that typically includes laterally-continuous, fine-texture confining units that can impede both recharge rates and regional propagation of increases in hydraulic head. Greater feasibility of MAR hinges on identifying locations where rapid, high-volume recharge that provides regional increases in pressure head are possible, but relatively little research has evaluated the factors that control MAR feasibility in alluvial groundwater basins. Here, we combine a transition probability Markov-chain geostatistical model
of the subsurface geologic heterogeneity of the east side of the northern Central Valley, California, with the 3D, variably-saturated water flow code, ParFlow, to explore the variability of MAR feasibility in this region. We use a combination of computationally-efficient local and global sensitivity analyses to evaluate the relative importance of factors that contribute to MAR feasibility. A novel proxy parameter approach was used to describe the configuration and proportions of subsurface hydrofacies and water table depth for sensitivity analyses, and results suggest that recharge potential is relatively more sensitive to
the variability of this proxy parameter than to the variablity of individual hydrofacies hydraulic properties. Results demonstrate that large variability of MAR feasibility is typical for alluvial aquifer systems and that outsize recharge rates are possible in select locations where interconnected, coarse-texture hydrofacies occur.



# 1 Introduction

Geologic heterogeneity strongly affects both the movement of water in the subsurface and the exchange of water between
subsurface and surface stores; however, rarely are enough data available to explicitly represent heterogeneous geologic features
in groundwater models (Koltermann and Gorelick, 1996; De Marsily et al., 2005). Instead, models typically simplify and/or
upscale heterogeneity to represent subsurface flows for purposes of regional-scale water resources management (e.g., Fogg,
1986; Phillips and Belitz, 1991). Upscaling methods have been the focus of numerous studies (e.g., Renard and De Marsily,
1997; Fogg et al., 2000; Neuman and Di Federico, 2003; Fleckenstein and Fogg, 2008), and coarse-resolution models with
upscaled (i.e., effective) hydrologic properties are often adequate for regional-scale flow studies, but typically lack enough
detail to reliably capture some phenomena, like recharge and transport processes, that are strongly influenced by geologic
heterogeneity.

To represent the influence of geologic heterogeneity on flow and transport phenomena, many approaches have relied on
stochastic methods, like transition probability based indicator geostatistics which can represent heterogeneous features while
honoring measured data (Carle and Fogg, 1996; Weissmann and Fogg, 1999; Weissmann et al., 1999). These approaches
represent geologic heterogeneity with hydrogeologic facies categories, each of which is assigned effective values or probability
densities for estimates of hydraulic properties. By categorizing facies according to depositional environment rather than texture
alone, the predictable geometries (i.e., facies mean lengths, proportions, and juxtapositions) of these features can be more
accurately represented with sparse data. Studies that rely on these methods show strong influence of subsurface heterogeneity
on groundwater/surface-water interactions and recharge processes (Lee, 2004; Fleckenstein et al., 2006; Engdahl et al., 2010;
Liu, 2014), including managed aquifer recharge (MAR) (Maples et al., 2019), especially for instances when the mean lengths
and proportions of high-permeability facies allow for percolation, i.e., formation of connected networks (Fogg et al., 2000;
Harter, 2005).

Accurately assigning aquifer properties in models can be a challenge because they are scale dependent attributes that are
challenging to measure and can vary over many orders of magnitude in typical aquifer systems (e.g., Sudicky, 1986; Gelhar
et al., 1992; Weissmann and Fogg, 1999). While aquifer tests can accurately constrain estimates of hydraulic conductivity ($K$)
for high-permeability facies, they are typically unreliable for estimating $K$ of low-permeability (i.e., aquitard) facies (Fogg,
1986; Fogg et al., 1998), which have been shown to influence pumping response (Fogg et al., 2000) and be important for
accommodating recharge (Maples et al., 2019). Reconciling typically sparse measurements of aquifer properties from aquifer
tests with the representation of effective values in models is often the source of large uncertainty because parameterization of
the properties in models is scale dependent (Sudicky and Huyakorn, 1991), and is typically achieved through model calibration.

The contaminant transport community has long recognized the strong influence of $K$ scaling and geologic heterogeneity on
transport processes (e.g., Gelhar et al., 1992; Sudicky and Huyakorn, 1991; Koltermann and Gorelick, 1996), and recent work
has extended these concepts to assess their role on runoff generation, evapotranspiration (ET), and feedbacks between subsur-
face and land-surface water in integrated hydrologic models (Srivastava et al., 2014; Gilbert et al., 2016; Foster and Maxwell,
2019), but relatively little research has focused on the influence these factors for MAR processes specifically. Recent work has





highlighted the importance of connected networks of high-$K$ facies for MAR (Maples et al., 2019), but to our knowledge the sensitivity of MAR processes to these heterogeneous geologic features as compared to other uncertain hydraulic properties has not been formally evaluated. Here, we simulate variably-saturated MAR dynamics in a highly-resolved representation of

complex subsurface geologic heterogeneity of a clastic, unconsolidated sedimentary aquifer system that includes both interconnected, high-$K$ sand and gravel deposits intermingled with silt- and clay-dominated sediments. We use a combination of local and global sensitivity analyses to provide insight into the relative importance of the subsurface geologic facies configuration and parameterization of subsurface hydraulic properties on MAR processes. This work provides insight into important factors to consider when investigating potential MAR sites and also highlights the utility of a combination of computationally-frugal

local and global sensitivity analyses for computationally-intensive hydrologic models.

## 2    Materials and Methods

### 2.1    Local Hydrogeology and Domain Extent

The model domain covers about 1640m$^2$ of the east side of the northern Central Valley, California, near the convergence of the lower portions of the American and Cosumnes Rivers with the Sacramento River (Fig. 1). The domain comprises a low-angle

alluvial fan complex that is typical of the Central Valley where previous studies have documented the presence of deposits that are favorable for recharge (Shlemon, 1967; Meirovitz, 2010), including massive, interconnected, highly-permeable sand and gravel deposits known as incised valley fill [IVF, (Weissmann et al., 2004, 2005)] that form from river incision and deposition events during cyclic Plio-Pleistocene Sierra Nevada glaciation. In places, multiple IVF deposits have been shown to overlap and interconnect from land surface into the deeper aquifer system, forming massive, coarse-texture, relatively high-permeability

pathways for recharge that bypass local, otherwise laterally-extensive confining units. These features have been shown to accommodate recharge volumes that are orders of magnitude greater than would be possible over the rest of the landscape (Maples et al., 2019). Other studies have shown that IVF features likely occur on river fans throughout the Central Valley (Weissmann et al., 2005), and in similar glacially-influenced rivers (Pierce and Scott, 1983) but are still largely undocumented.

The local hydrostratigraphy of the area is described in detail by Meirovitz (2010) and Maples et al. (2019). In general, the two

major rivers intersecting the domain, the American and Cosumnes Rivers, have markedly different depositional characteristics. The American River drains a large (>4000 km$^2$), high-elevation catchment that extends to the Sierra Nevada crest [>3000 m above mean sea level (amsl)]. As a result, the American River was greatly influenced by cyclic plio-pleistocene glaciation that deposited IVF in the domain area. Conversely, the Cosumnes River catchment is smaller (900 km$^2$) and lower in elevation. As a result, deposits from the Cosumnes River do not contain IVF and are typically finer in texture. In some locations in the domain

area, Quaternary and Holocene channel avulsion of the American River resulted in a more southwest course that intersects the current path of the Cosumnes river, creating complex overlapping stratigraphy in that area. Cross-cutting IVF and overlapping paleochannel networks in the domain area result in an aquifer system that is typically unconfined (and sometimes perched) or semi-confined at shallow depths and increasingly confined with depth (Fleckenstein et al., 2006; Liu, 2014; Niswonger and





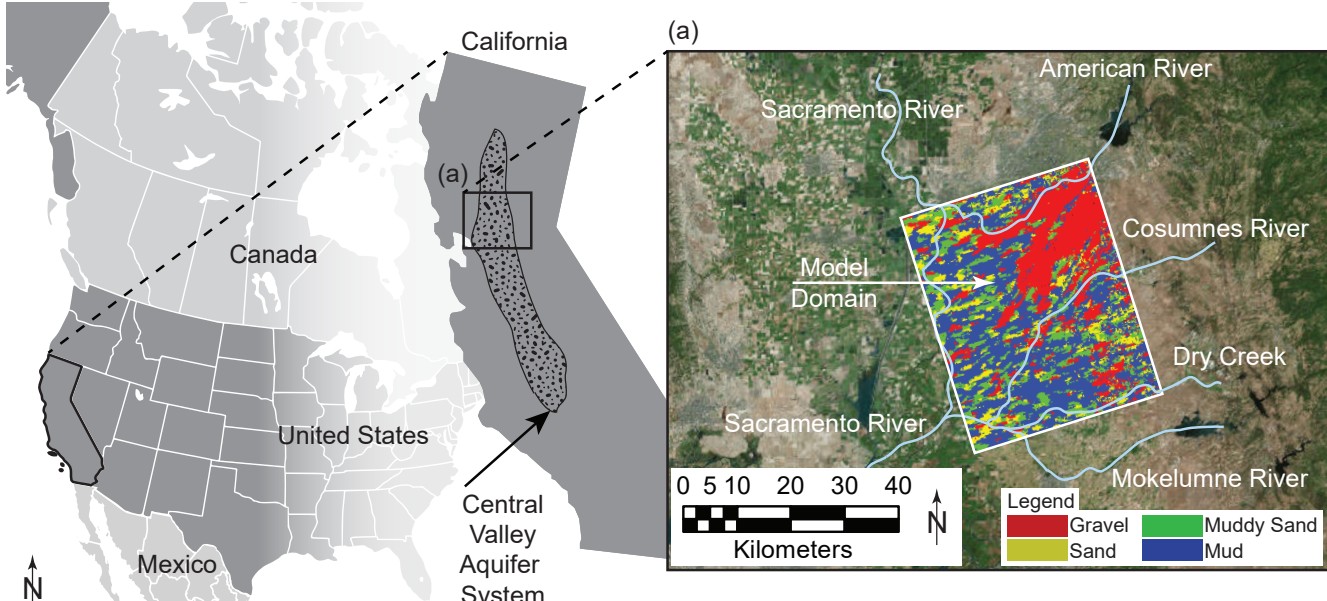

**Figure 1.** Location of model domain in the in the Central Valley aquifer system in California, and (a) inset of the uppermost layer of the model hydrofacies overlain over aerial imagery of the Central Valley and local river systems. *(double-column width)*

Fogg, 2008). Groundwater pumping in the region typically occurs at depths >30 m in the deeper semi-confined or confined

portion of the aquifer system (Liu, 2014).

## 2.2    Hydrofacies Model Development

Transition probability Markov-chain geostatistics (TPROGS) (Carle and Fogg, 1996, 1997; Carle, 1999), was used to simulate the subsurface distribution of hydrofacies in the domain area (Fig. 2). Model development is described in detail by Meirovitz (2010) and Maples et al. (2019). Conditioning data for the TPROGS model included about 1200 well logs, soil surveys,

geologic cross-sections and mapped paleochannels. Geologic data were binned into four textural categories: gravel, sand, muddy sand, and mud (undifferentiated silt and/or clay) [Table 1 (Fleckenstein et al., 2004; Meirovitz, 2010)]. "Mud" refers to silt and clay undifferentiated, because most of the subsurface data available only identify the fine-grained sediments and are not sufficiently detailed to distinguish silt from clay. From these data the proportions for each facies were calculated directly. Through geologic analysis of the data, additional parameters were estimated describing the mean lengths of each hydrofacies

along the principal directions and the embedded transition probabilities to represent cross-correlation between different facies. Because the depositional characteristics of the American and Cosumnes fans were markedly different, individual models of each were produced and subsequently combined by Meirovitz (2010).





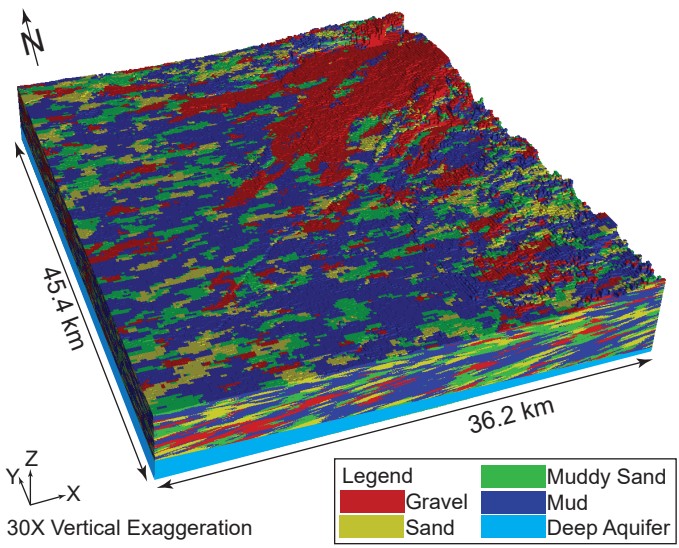

**Figure 2.** Three-dimensional representation of the model domain (Meirovitz, 2010; Liu, 2014).

*TABLE 1 ABOUT HERE.*

The model uses an orthogonal grid geometry with 181 × 227 × 265 cells in the x-, y-, and z- directions, respectively. The x- and y- directions of the grid were rotated 17.85 degrees counterclockwise from the cardinal directions, and the z-direction was oriented vertically. Cell sizes were 200 × 200 × 1 m. The total domain size is 36.2 × 45.4 × 0.265 km, in the x-, y- and z-directions, respectively. Cells located above land surface were designated as inactive in the model, resulting in about 7.3 million active cells in the domain area.

## 2.3 Hydrologic Model Development

### 2.3.1 Governing Equations

Three-dimensional, variably-saturated water flow was simulated with the hydrologic modeling code, ParFlow (Ashby and Falgout, 1996; Jones and Woodward, 2001; Kollet and Maxwell, 2006), which couples surface and subsurface flow with the 2-D diffusive or kinematic wave equation, and solves the 3-D mixed form of Richards' Equation for variably-saturated subsurface flow:

$$S_s S_w(h)\frac{\partial h}{\partial t} + \phi\frac{\partial S_w(h)}{\partial t} = \nabla \cdot \mathbf{q} + q_r(x,z) \tag{1}$$

where

$$\mathbf{q} = \phi S_w(h)\mathbf{v} = -\mathbf{K}_s(x)k_r(h)\nabla(h+z) \tag{2}$$





In these equations $S_s$ is specific storage [L$^{-1}$], $S_w$ is relative saturation [$-$], $h$ is pressure head [L], $t$ is time [T], $\phi$ is porosity [$-$], $\mathbf{q}$ is Darcy flux [L T$^{-1}$], $q_r$ is a source/sink term [T$^{-1}$], $z$ is elevation [L], $\mathbf{v}$ is the subsurface flow veolicty [L T$^{-1}$],

$\mathbf{K}_s(\mathbf{x})$ is the saturated hydraulic conductivity tensor [L T$^{-1}$], and $k_r$ is relative permeability [$-$]. The van Genuchten relations (Van Genuchten, 1980) describe $S_w$ and $k_r$ as a function of $h$ in the unsaturated zone, with parameters for air entry pressure $\alpha$ [L$^{-1}$], pore size distribution $n$ [$-$], and residual saturation $S_{res}$ [$-$].

### 2.3.2 Boundary Conditions

Model boundary conditions are discussed in greater detail in Liu (2014) and Maples et al. (2019). The locations of domain
boundaries were chosen to simplify the assignment of boundary conditions for the flow model. The eastern boundary roughly coincides with the Sierra Nevada foothills, and the northern, southern, and western boundaries roughly coincide with local surface water bodies (Fig. 1). A specified head boundary condition was applied for the eastern boundary to coincide with the local groundwater head distribution estimated from local monitoring well data (Liu, 2014). A general head boundary of 0 m amsl was set 1 km beyond the western boundary to approximate the Sacramento River and Sacramento-San Joaquin Delta
along the northwestern, and southwestern portions of the western boundary, respectively. No-flow boundary conditions were applied along northern, southern, and bottom boundaries because the regional groundwater flow direction is generally from east to west. Model spin up and recharge simulations used combinations of specified-flux and specified-head upper boundary conditions and are described in greater detail in subsequent sections.

### 2.3.3 Model Spinup and Calibration

Model spinup and calibration are described in greater detail in Liu (2014) and Maples et al. (2019). To summarize, a 16-yr. simulation period was used to bring the simulated hydrology into dynamic equilibrium. Water budget components, including groundwater discharge, recharge and boundary flows along with facies hydraulic properties were estimated and adjusted manually to simulate a realistic water budget, water table configuration, and vertical hydraulic gradients during the calibration process. An initial potentiometric surface was specified using interpolated groundwater level data. Monthly estimated urban
and agricultural groundwater pumping rates were applied as specified fluxes representing wells screened in lower portions of the domain that coincide with typical screened intervals of municipal and agricultural pumping wells in the region. Dominant sources of recharge for the region include stream recharge from the American River, Cosumnes River, and Deer Creek, as well as deep percolation from agricultural and urban return flows. Weekly estimates of spatially-distributed river stage for the streams were applied as specified heads along coincident land surface cells. Monthly estimates of urban and agricultural
recharge volumes were applied as specified-flux boundary condition across the top of the domain to simulate deep percolation of agricultural and urban return flows and to equilibrate soil moisture conditions in the near-surface UZ cells. Hydraulic properties for each facies category were calibrated manually (Table 2) and are consistent with the range of literature values for the Central Valley, California, and for similar alluvial systems (Anderson et al., 2015; Botros et al., 2009; Fleckenstein et al., 2004; Frei et al., 2009; Maserjian, 1993; Niswonger and Fogg, 2008; Sager, 2012).


*TABLE 2 ABOUT HERE.*

All simulations were performed using the Cheyenne high-performance cluster at NCAR's Computational and Information Systems Laboratory (doi:10.5065/D6RX99HX). The numerical problem was distributed on 540 cores for each simulation. Approximately 450 model evaluations were required for the exploratory simulations, local sensitivity analyses, and global sensitivity analyses described in subsequent sections, which required approximately 400,000 core-hours of computing time in total. The large computational expense for each simulation (890 core-hours per simulation, on average) required that computationally resources be allocated efficiently.

## 2.4 Exploratory Simulations

### 2.4.1 Site Selection

One hundred 1 km$^2$ recharge sites, each encompassing 25 upper-boundary cells, were chosen to approximate hypothetical MAR infiltration basins (Fig. 3a). Each site was randomly selected from a 910 km$^2$ region within the domain that excluded locations within 5 km of lateral domain boundaries to minimize the influence of boundary conditions. The 100 exploratory sites encompass roughly 6% of the total domain area, which was deemed sufficient to sample the variability of site characteristics observed across the domain. The size of each site was chosen to reflect a regional-scale MAR site, which range from large networks of basins >25 km$^2$ in size (e.g., Kern Water Bank Authority, 2018) to individual infiltration basins over several hectares or smaller (e.g., Beganskas and Fisher, 2017) in California.

### 2.4.2 Site Characteristics

Maples et al. (2019) highlighted that the (1) relative proportions, and degree of vertical interconnection, of coarse-texture facies (sand and gravel) and (2) the unsaturated-zone thickness beneath recharge sites are important factors for recharge feasibility. In this study, we sought to develop site characteristics to describe recharge feasibility across the domain, including approximations of how effective (i.e., upscaled) vertical $K_s$ varies spatially. Here, we use a heuristic approach of simple averages to bound the expected range of upscaled vertical $K_S$, where the arithmetic and harmonic mean ($K_{arith}$ and $K_{harm}$) are the upper and lower bounds, respectively, and the geometric mean ($K_{geom}$) is an intermediate value. $K_{arith}$ and $K_{harm}$ are typically used to approximate groundwater flow parallel and perpendicular to layering, respectively, in anisotropic systems (Freeze and Cherry, 1979). This concept has been generally been extended to variably-saturated flow (Mualem, 1984; Assouline and Or, 2006). Fogg et al. (2000) showed that vertical groundwater flow in systems with vertically-connected networks of permeable facies tends toward values between $K_{arith}$ and $K_{geom}$. Values for $K_{arith}$, $K_{geom}$, and $K_{harm}$ for $n$ vertically-coincident cells are given as:

$$K_{arith} = \frac{K_1 + K_2 + ... + K_n}{n} \tag{3}$$



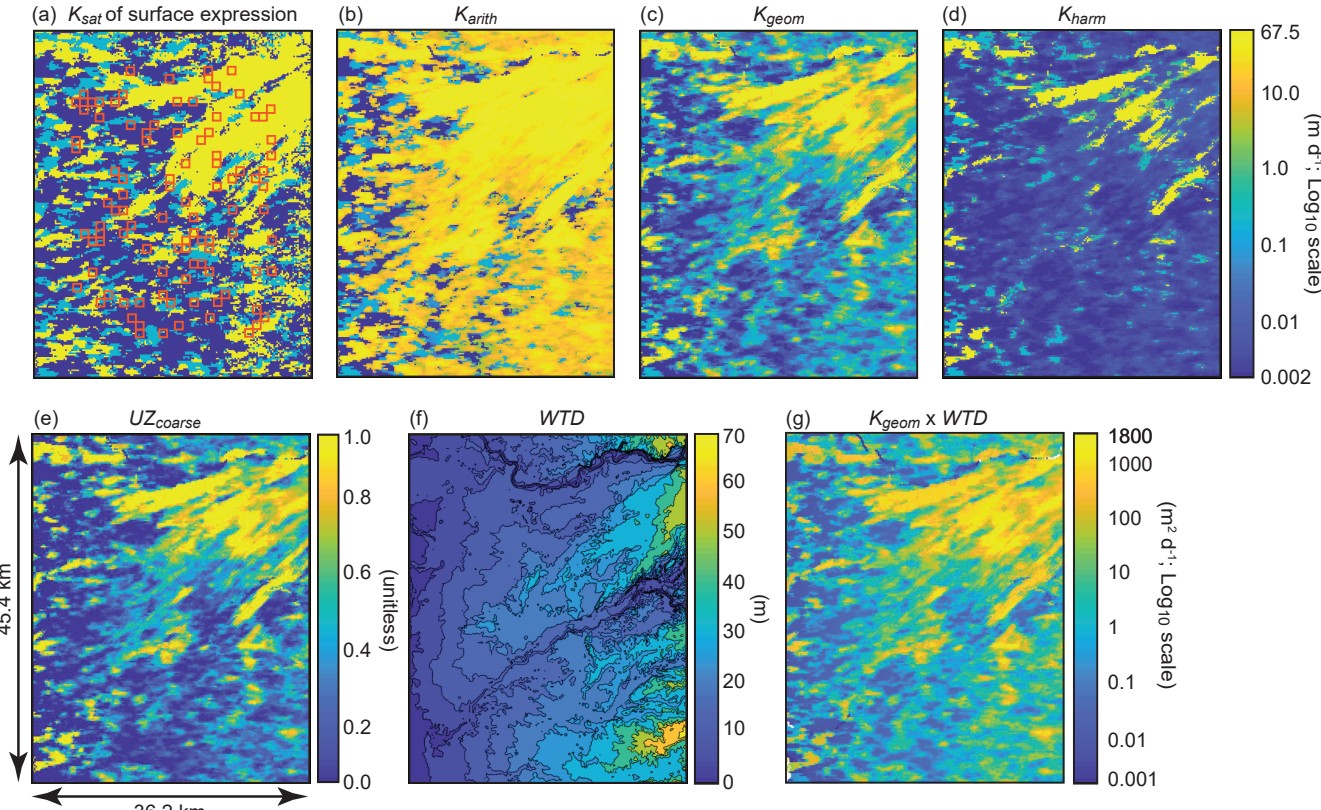

**Figure 3.** Plan view of the model domain, where the first row shows (a) $K_{sat}$ of the uppermost model layer (i.e., the surface expression) overlaid with the locations of 100 randomly-sampled 1-km$^2$ exploratory recharge sites (red squares) randomly chosen from 910 potential locations, along with (b) arithmetic, (c) geometric, and (d) harmonic mean of vertical $K_{sat}$ for unsaturated zone (UZ) facies (i.e., $K_{geom}$, $K_{arith}$, $K_{harm}$, respectively). The second row shows (e) the coarse-texture (gravel and sand) fraction of UZ facies ($UZ_{coarse}$), (f) simulated initial depth-to-water ($WTD$), and (g) $K_{geom}$ multiplied by $WTD$ ($K_{geom} \times WTD$). Locations >5 km from the lateral domain boundaries were excluded from the potential sites to avoid interference with boundary conditions. *(double-column width)*

$$K_{geom} = \sqrt[n]{K_1 \times K_2 \times ... \times K_n} \tag{4}$$

$$K_{harm} = \frac{n}{\frac{1}{K_1} + \frac{1}{K_2} + ... + \frac{1}{K_n}} \tag{5}$$


Site characteristics are described in detail in Table 3 and include upscaled $K_s$ approximations $K_{arith}$, $K_{geom}$, and $K_{harm}$, along with initial unsaturated-zone thickness ($WTD$), proportion of coarse-texture sand and gravel unsaturated-zone facies





($UZ_{\text{coarse}}$), and proportion of coarse-texture facies at land surface ($Surf_{\text{coarse}}$). Each site characteristic was calculated for a control volume encompassing vertically-coincident cells from the land surface to the initial water table depth for each site. For

the purposes of this work, the interface of the deeper aquifer system was designated as the initial water table depth ($WTD$). Additional site characteristics were developed by combining existing site characteristics, i.e., $WTD$ was used as a multiplier for $K_{\text{arith}}$, $K_{\text{geom}}$, $K_{\text{harm}}$, $Surf_{\text{coarse}}$, and $UZ_{\text{coarse}}$. Spatial distributions of select site characteristics are shown in Fig. 3.

*TABLE 3 ABOUT HERE.*

Each site was further evaluated according to whether there was vertical connectivity (i.e., percolation) of coarse-texture

facies from land surface into the deeper aquifer system. Interconnection was defined within each site control volume by a 6-connectivity metric (Pardo-Iguzquiza and Dowd, 2003), in which neighboring gravel and sand cells are said to be connected if they intersect along a face. Coarse-texture facies were said to percolate if any combination of gravel and sand facies were interconnected vertically within the control volume from land-surface to the initial water table.

### 2.4.3 Recharge Scenarios and Model Post-Processing

To evaluate the system response to recharge stress and recovery, 90-day recharge scenarios were run individually at each site, wherein recharge was simulated during the initial 30-day period followed by a 60-day recovery period. Surface ponding was approximated by a specified head boundary condition representing 10 cm of ponding at the 25 upper-boundary cells coincident with each recharge site, with no recharge specified for the remaining upper-boundary cells. An additional simulation was run in which no recharge was specified for all upper-boundary cells, i.e., as a no-recharge scenario. The initial condition for all

recharge and no-recharge scenarios was the $h$ distribution from the end of the model spin up. Results were output at 5-day intervals for all simulations.

Following Maples et al. (2019), recharge responses were isolated from other model stimuli by differencing $h$, and total subsurface water storage ($TSS$) from each colocated cell at each timestep, in each recharge and no-recharge simulation. In this way, perturbations in $h$ and $TSS$ result from the recharge stress alone, while other stimuli, including transient model

response to regional boundary condition effects, are eliminated. For each simulation, $h$ perturbations were evaluated at a 10-cm threshold. Domain-wide perturbations in $h$ and $TSS$ from recharge stress were evaluated for all 100 recharge simulations at $t = 30$ days to calculate the volumetric extent of subsurface pressure perturbation, $P_{\text{30d}}$ [L$^3$], and the effective recharge rate, $R_{\text{30d}}$ [L T$^{-1}$], respectively, at the end of the 30-day recharge stress period for each site. Similarly, domain-wide perturbations in $TSS$ were evaluated at $t = 90$ days and were further delineated according to whether the change-in-storage occurred in fine-

texture (muddy sand and mud) or coarse-texture (sand and gravel) facies, so that the proportion of the total recharge volume accommodated by fine-texture facies, $V_{\text{fines, 90d}}$ [$-$], could be evaluated. Previous work has highlighted the importance of fine-texture facies for accommodating recharge, especially during late time (e.g., Maples et al., 2019). All model outputs used for subsequent analyses are shown in Table 3.





### 2.4.4 Relations between Site Characteristics and Recharge Potential

To better understand the relationships between site characteristics (variables) and model outputs (predictors), correlations (Pearson's $r$, Spearman's rho, and Kendall's tau) were evaluated between all variable and predictor pairs across all 100 sites. Variables and predictors are described in Table 3. The purpose of evaluating correlations between variables and predictors was to determine if any site characteristics could be used to reasonably predict model outputs with empirical relations. $Log_{10}$ data transformations were selectively performed on variables and predictors to improve normality prior to calculation of Pearson's

$r$. Transformations were not performed for Spearman's rho and Kendall's tau, as neither require normal distributions for prediction.

### 2.4.5 Development of a Geologic Proxy Parameter for Recharge Potential

To incorporate descriptions of geologic configuration in sensitivity analyses of recharge potential, development of a geologic proxy parameter ($GPP$) was required. Correlations between select variables ($R_{30d}$, $P_{30d}$, and $V_{fines, 90d}$) and predictor pairs

described in section 2.4.4 were ranked, and a $GPP$ was determined for each by developing empirical regression relations between those variables and the highest ranked predictor. Unlike other parameters, the $GPP$ cannot be directly varied at each site, and instead was approximated either with linear regression of recharge response and $GPP$, or by relocating the recharge site within the domain to a location with the corresponding $GPP$. Previous studies have shown the importance of geologic heterogeneity on MAR feasibility (e.g., Maples et al., 2019), and the novel proxy parameter methodology described here is

analogous to a transfer function (e.g., Wösten et al., 2001), in that it describes the influence of complex geologic heterogeneity on recharge processes with relatively easily-derived site characteristics. By using this approach in sensitivity analyses, we are able to both capture this geologic complexity and also reduce the overall computational expense, albeit with some predictive uncertainty related to the regression relations.

### 2.5 Sensitivity Analyses using Fit-Independent Statistics

Realistic ranges of model parameters describing eight hydraulic properties for four facies types ($n = 24$ model parameters) are shown in Table 4 along with an estimated range of $GPP$. Ranges for model parameters were chosen from literature values for the Central Valley California, and for similar alluvial systems (Anderson et al., 2015; Botros et al., 2009; Fleckenstein et al., 2004; Frei et al., 2009; Maserjian, 1993; Niswonger and Fogg, 2008; Sager, 2012). Model parameters were assumed to be distributed uniformly within each of these ranges for simplicity. The range of $GPP$ was determined from the range observed

from the 100 exploratory sites described in section 2.4. The distribution $GPP$ was observed to be approximately log-normal, so a $Log_{10}$ data transformation was performed for subsequent sensitivity analyses.

*TABLE 4 ABOUT HERE.*





All sensitivity scenarios were initialized with the $h$ distribution from the end of the model spin up and were simulated and post-processed following the approach outlined in section 2.4.3. In this way, each scenario required two simulations be run with
the same parameter sets (i.e., a recharge and no-recharge simulation) which were then differenced to isolate recharge stresses from other model stimuli, including transient model responses to changes in parameter values.

### 2.5.1 Local Sensitivity Analyses

Parameter sensitivities were evaluated locally using the dimensionless-scaled sensitivity ($DSS$) and composite-scaled sensitivity ($CSS$) metrics, which are computationally-frugal screening methods used to compare the relative importance of different
parameters to the estimation of a simulated model output (Hill and Tiedeman, 2007). $DSS$ for simulated output $i$ and parameter $j$ are calculated as

$$DSS_{ij} = \left( \frac{\partial y_i^{'}}{\partial b_j} \right)\Bigg|_{\mathbf{b}} |b_j| \omega_{ii}^{1/2} \tag{6}$$

where $y_i^{'}$ is the $i$th simulated output, $b_j$ is the $j$th estimated parameter, $\partial y_i^{'}/\partial b_j$ is the derivative (i.e., the sensitivity) of the simulated output with respect to the $j$th parameter, $\mathbf{b}$ is the vector of parameter values at which sensitivities are evaluated, and
$\omega_{ii}^{1/2}$ is the weight of the $i$th simulated output. For this work, simulated outputs $P_{30d}$, $R_{30d}$, $V_{\text{fines, 90d}}$ were weighted equally at unity.

Composite-scaled sensitivities ($CSS$) were calculated to estimate the total amount of sensitivity provided by each parameter across multiple sites and for multiple model outputs

$$CSS_j = \sum_{i=1}^{n} \left[ (DSS_{ij})^2 \big|_{\mathbf{b}} / n \right]^{1/2} \tag{7}$$

where $DSS_{ij}$ is from equation 6, and $n$ is the total number of simulated outputs $i$ associated with parameter $j$.

$DSS$ were estimated for select model outputs $R_{30d}$, $P_{30d}$ and $V_{\text{fines, 90d}}$ (Table 3) by perturbing each hydraulic-property parameter ($n = 24$) by 10% of its total range (Table 4). Results from the exploratory simulations show that recharge response is highly dependent on site choice, so $DSS$ was evaluated at four representative sites which span a large range of recharge potential. Each of the four representative sites were chosen to correspond with the 25th, 50th, 75th and 95th percentile of
recharge potential, as estimated by $GPP$. These sites are hereto referred to as q25, q50, q75, and q95, respectively. A total of 96 model evaluations were required to estimate $DSS$ on the three model outputs ($R_{30d}$, $P_{30d}$ and $V_{\text{fines, 90d}}$) for all 24 parameters (Table 4) at these four sites. To incorporate $GPP$ in $DSS$ analyses, an approach was developed using the predictive regression relation between $GPP$ and $R_{30d}$. For example, $DSS$ require perturbation of a parameter (i.e., $\partial b_j$) by a percentage of the parameter range (e.g., by 10%). A corresponding 10% perturbation in $R_{30d}$ (i.e., $\partial y_i^{'}$) was approximated using the predictive
regression relation between $GPP$ and $R_{30d}$ rather than by performing an additional model evaluation.





$CSS$ were calculated for $R_{30d}$, $P_{30d}$ and $V_{\text{fines, 90d}}$ (Table 3 by combining $DSS$ estimates for each model output across sites q25, q50, q75, and q95 for each of 24 model parameters. The same approach was used to calculated $CSS$ for $GPP$, but was only estimated for $R_{30d}$ and not for $P_{30d}$ and $V_{\text{fines, 90d}}$.

### 2.5.2 Global Sensitivity Analyses

A measure of global sensitivity was provided by the method of Morris (1991), which relies on the calculation of elementary effects, i.e., local derivatives sampled one-at-a-time (OAT) on a grid that covers the parameter space. The method of Morris creates a trajectory through the parameter space by perturbing each parameter $x_j$ along a grid by a step $\Delta_j$. A sequence of $p$ perturbations is required to obtain a one trajectory for a model with $p$ parameters. For each trajectory, the elementary effect for a single parameter, $EE_j$, is calculated as the ratio of the perturbation in model output to the perturbation of the parameter

$$EE_j = \frac{f(x_1,...,x_j+\Delta_j,...,x_p) - f(x)}{\Delta_j} \tag{8}$$

where $f(x)$ is the evaluation of the function at the prior point in the trajectory. Calculating the elementary effects for $p$ parameters using a single trajectory requires $p+1$ model evaluations. Because the elementary effect for any single trajectory does not account for interactions between parameters and depends strongly on the location of the initial point, $x$, in the parameter space, the method of Morris performs the OAT approach over multiple trajectories, $N$, within the parameter space using a factorial

sampling approach. A variation of the original approach was employed to resolve issues related to opposite signs of elementary effects affecting the calculation of total-order sensitivity (Campolongo et al., 2007), in which the total-order sensitivity of each parameter, $\mu_j^*$, was calculated as the mean of the absolute values of $N$ elementary effects

$$\mu_j^* = \frac{1}{N} \sum_{k=1}^{N} \left| EE_j^k \right| \tag{9}$$

Unlike $DSS$ approaches, the method of Morris required significantly greater computational resources for an equivalent

number of parameters, so a sub-set of three parameters were included in the Morris approach. Results from local sensitivity analyses suggest that $K_s$ and $GPP$ are relatively important parameters for recharge potential. Computational expense was further reduced by pairing parameters, i.e., $K_s$ of gravel and sand, and of muddy-sand and mud, effectively reducing these four parameters into two describing 'coarse-' and 'fine-texture' facies, respectively. By pairing parameters, $K_s$ of gravel and sand (and of muddy sand and mud) are perturbed within their respective parameter ranges together, reducing the total number

of parameters from five to three. Sensitivity indices were calculated using a sample size of $N = 20$, resulting in a total of 80 model evaluations. Herman et al. (2013) demonstrated that the method of Morris with $N = 20$ trajectories produced similar sensitivity results to the Sobol' method (Sobol, 2001) with >2 orders-of-magnitude fewer model evaluations.

To further reduce the computational expense, the total simulation time was reduced from 90 to 10 days, during which recharge was applied for the entire simulation. Sensitivity indices, Morris $\mu_j^*$, were only evaluated with respect to the effective





recharge rate at the end the 10-day simulation period, $R_{10d}$ [cm d$^{-1}$]. Morris $\mu_j^*$ was not evaluated with respect to other model outputs describing pressure perturbation or volume of recharge accommodated by fines because $GPP$ was determined to be an inadequate predictor of these model outputs.

To incorporate $GPP$ in the Morris framework, a novel approach was developed in which the location of the sampling site was varied to correspond with the requisite $GPP$ parameter choice. For example, if a hypothetical sensitivity analysis required

evaluation of the model with $GPP$ at the 50th quantile (q50; i.e., the median value), the model would be run using the site with the nearest corresponding $GPP$ value from the 100 exploratory sites described in section 2.4. In this way, the variability of $GPP$ as identified in the exploratory simulations can be sampled directly by simply varying the location of the recharge site within the domain. The Morris approach was implemented with open-source library developed by Herman and Usher (2017).

## 3    Results and Discussion

### 3.1    Exploratory Simulations

Results from the exploratory simulations at the 100 selected sites show a wide range of $R_{30d}$, $P_{30d}$, and $V_{fines, 90d}$ across sites (Fig. 4). $R_{30d}$ varied over 2 orders of magnitude and were non-normally distributed, with a maximum, minimum, and mean value of 66.4, 0.5, and 8.6 cm d$^{-1}$. $P_{30d}$ were similarly non-normally distributed and also showed a large range, varying over 4 orders of magnitude. Maximum, minimum, and mean $P_{30d}$ were $1.6 \times 10^5$, 33, and $1.9 \times 10^3$ m$^3$, respectively. These results

highlight that a small number of sites have outsize recharge potential compared with most of the landscape. $R_{30d}$ and $P_{30d}$ were positively correlated ($r > 0.70$), indicating that these recharge benefits are physically related. The proportion of recharge accommodated by fine-texture facies ($V_{fines, 90d}$) also showed large variability across sites, ranging from 0.13 to 1.00, with a mean value of 0.69. The high proportion of $V_{fines, 90d}$ observed here is consistent with previous findings that suggest that fine-texture facies are the largest reservoir for MAR in this aquifer system (Maples et al., 2019). $V_{fines, 90d}$ was negatively correlated

with both $R_{30d}$ and $P_{30d}$ ($r > 0.70$), which indicates that when interconnected, coarse-texture pathways are present, a greater proportion of MAR is accommodated in the coarse-texture aquifer system.

### 3.1.1    Influence of Coarse-Texture Connectivity

Of the 100 exploratory sites, 23 were shown to have interconnected coarse-texture gravel and sand facies from land surface to the initial water table depth. $R_{30d}$, $P_{30d}$, and $V_{fines, 90d}$ were parsed according to whether they were interconnected (Fig.

4). Results show that mean $R_{30d}$ and $P_{30d}$ were 2.2× and 2.3× greater, respectively, for interconnected sites than for non-interconnected sites (14.7 vs 6.7 cm d$^{-1}$, and $1.6 \times 10^4$ vs. $6.9 \times 10^3$ m$^3$, respectively). Mean $V_{fines, 90d}$ were 1.3× greater for non-interconnected sites than for interconnected sites. Distributions of $R_{30d}$ and $P_{30d}$ for interconnected and non-interconnected sites differed significantly according to the two-sample Kolmogorov-Smirnov test. Interconnected and non-interconnected distributions of $V_{fines, 90d}$ were not significantly different.

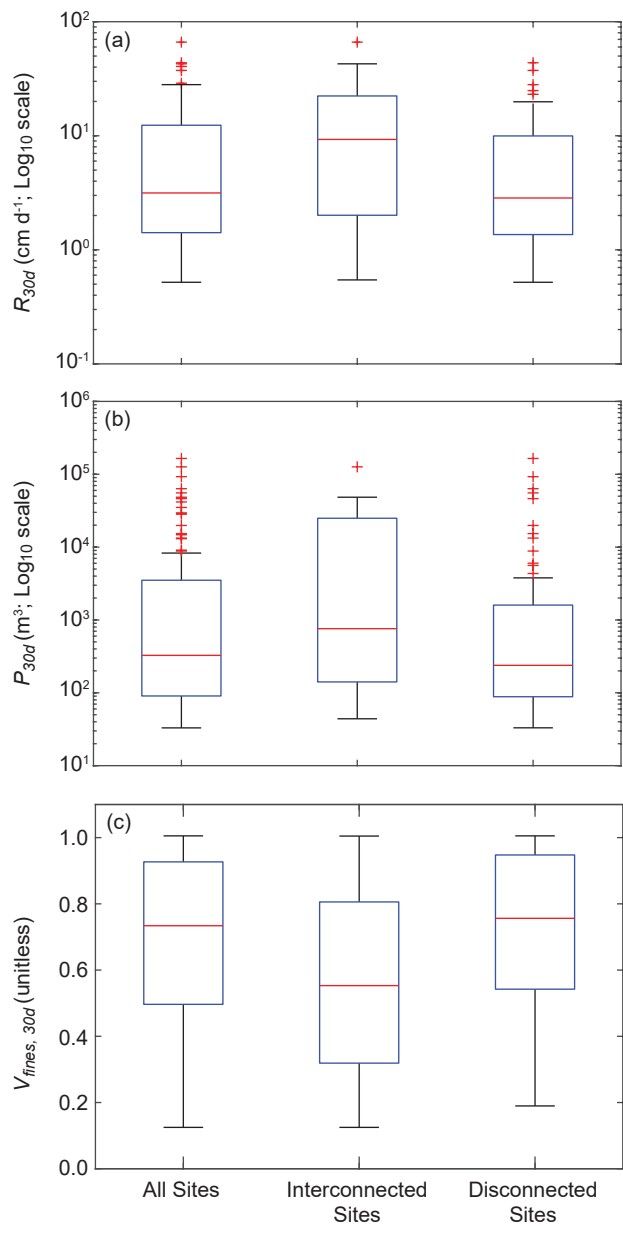

**Figure 4.** Box plots of the (a) 30-day average recharge rate, $R_{30d}$ and (b) 30-day pressure perturbation area of influence, $R_{30d}$ for all exploratory simulations ($n = 100$). Additionally, sites were parsed according to whether there was vertical interconnection of coarse-texture facies from land surface to the initial water table depth (i.e., interconnected sites, $n = 23$), or whether sites did not have interconnection of coarse-texture facies (i.e., disconnected sites, $n = 77$).





These results indicate that sites with interconnected coarse-texture facies have greater $R_{30d}$ and $P_{30d}$ potential. However, this metric is not entirely diagnostic of recharge potential. As shown in Fig. 4a,b, some interconnected sites exhibited low $R_{30d}$ and $P_{30d}$. This is likely because some interconnected sites with shallow water table depths have limited unsaturated pore volume to accommodate large recharge volumes. In addition, the interconnection metric described herein only describes vertical interconnection of coarse-texture facies for unsaturated-zone cells that are vertically coincident with the recharge

site, and does not consider whether these coarse facies connect with the greater aquifer network outside of the unsaturated-zone control volume. Results also show that some seemingly disconnected sites have large recharge potential. Indeed, the interconnection metric described here does not account for any lateral interconnection from land surface to the greater aquifer network, which could explain this behavior. In reality, the simplified estimator of connectivity used here likely underestimates the number of interconnected sites.

### 335 3.2 Recharge Metrics

#### 3.2.1 Correlation Matrices

A matrix of correlations (Pearson's $r$) of pairs of site characteristics and simulated outputs for the 100 exploratory simulations was generated to better understand the relationships between variables (Fig. 5). Strong correlation ($r > 0.70$) was observed for 6 of 52 pairs of site characteristics and simulated outputs. Strong correlation was also observed among many site characteristics and among the majority of simulated outputs (i.e, collinearity), which can make the choice of an optimal proxy parameter more

challenging. Site characteristics that include $K_{harm}$ were not shown in the correlation matrix because we were not able to improve normality of the distribution these data with a $\text{Log}_{10}$ data transformation; however, additional correlation metrics (Fig. 6) indicate that site characteristics that include $K_{harm}$ may also be strongly correlated.

#### 3.2.2 Ranked Correlations

Additional correlation metrics (Pearson's $r$, Spearman's rho, and Kendall's tau) between $R_{30d}$ and site characteristics were ranked and are shown in Fig. 6. Results show that site characteristics that include some variation of $K_{arith}$, $K_{geom}$, or $K_{harm}$ were, in general, more correlated with $R_{30d}$ than site characteristics that only include $WTD$, $UZ_{coarse}$, and $Surf_{coarse}$. $K_{geom} \times WTD$ was, on average, most correlated with $R_{30d}$.

  In general, site characteristics that included $K_{geom}$ and $K_{harm}$ were slightly more correlated with $R_{30d}$ than site characteristics

that included $K_{arith}$. We speculate that this behavior is related to the dominantly vertical flow direction of recharge across typically horizontal facies configurations. Previous work has shown that $K_{geom}$ and $K_{harm}$ best describe upscaled $K$ for these flow configurations in this domain (Yunjie Liu, personal communication; Fogg, 1986).

  Interestingly, site characteristics that included only $WTD$, $UZ_{coarse}$, and $Surf_{coarse}$ were poorly correlated ($r < 0.20$) with $R_{30d}$. This finding has important implications for determining MAR site suitability because many GIS-derived indices of

recharge suitability rely solely on soil and/or surface geology to determine geologic suitability for recharge. These results suggest that even more detailed geologic descriptions that estimate deeper fractions of coarse-texture facies may not fully





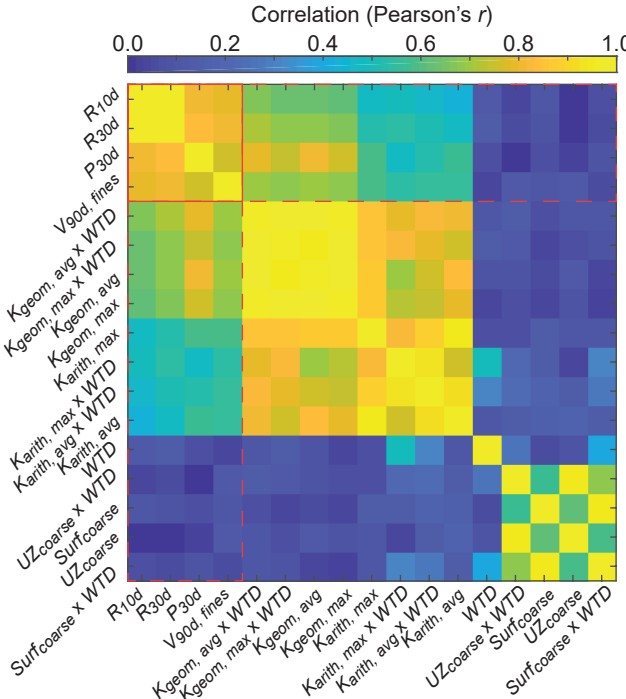

**Figure 5.** Correlations (Pearson's $r$) for all combinations of site characteristics and model outputs. Correlations among site characteristics are bounded by a solid red box, and correlations between site characteristics and model outputs are bounded by a dashed red box.

capture recharge potential. Instead, metrics that include some description of upscaled vertical $K$ appear to be most diagnostic of recharge potential.

### 3.3 Recharge Extrapolation

The relation between site-averaged $K_{\text{geom}} \times DTW$ and $R_{30d}$ was determined to be the best predictor and was used to predict $R_{30d}$ for subsequent sensitivity analyses by treating $K_{\text{geom}} \times DTW$ as a $GPP$ (Fig. 7a). The linear regression relation between $K_{\text{geom}} \times DTW$ and $R_{30d}$ was highly significant ($p < 0.01$), and correlation coefficients ($r^2$) showed that empirical regression explained 70% of the variation in the data. Linear regression relations for $K_{\text{geom}} \times DTW$ and $P_{30d}$ and $V_{\text{fines, 90d}}$ were deemed insufficient for prediction ($r^2 < 0.40$) and were not incorporated in sensitivity analyses.

Domain-wide $K_{\text{geom}} \times WTD$ was converted to $R_{30d}$ using the predictive relation described above (Fig. 8). Results show that 84% of the domain has $R_{30d}$ potential <10 cm d$^{-1}$, while 6% of the domain has $R_{30d}$ potential >25 cm d$^{-1}$, and a small portion of the domain has $R_{30d}$ potential >150 cm d$^{-1}$. These results show a large contrast between locations with high recharge potential and those with low recharge potential which supports previous findings indicating that a small fraction of the landscape has recharge potential that is orders-of-magnitude greater than the rest of the landscape (Maples et al., 2019; Fleckenstein et al.,





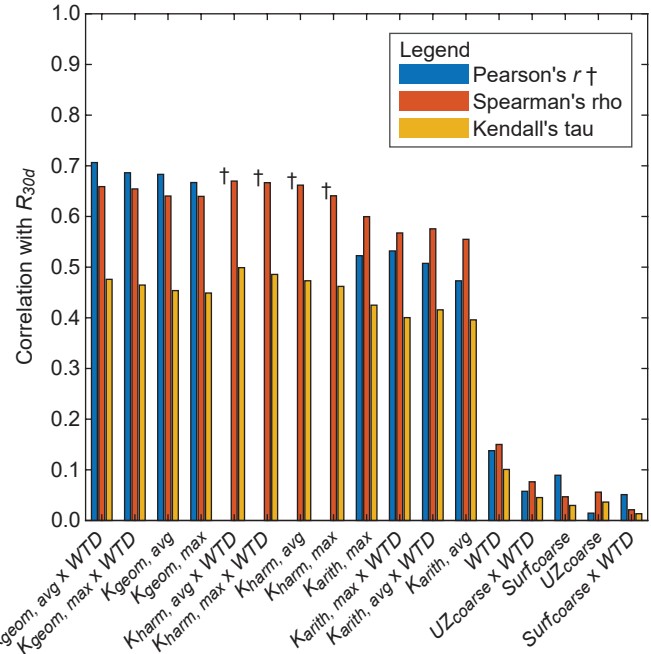

**Figure 6.** Ranked correlations (Pearson's $r$, Spearman's Rho, and Kendall's Tau) of site characteristics with 30-day average recharge rate ($R_{30d}$). †Pearson's $r$ was not evaluated for site characteristics where the normality of the distribution could not be improved with a $Log_{10}$ data transformation.

2006). Deposition of IVF within the domain area has been documented by Meirovitz (2010) and explains the presence of these high recharge potential locations.

### 3.4 Sensitivity Analyses

### 3.4.1 Local Sensitivity Analyses

$GPP$ perturbations to estimate $DSS$ for sites q25, q50, q75, and q95 using predictive regression relations are are illustrated

in Fig. 7b,c. $DSS$ and $CSS$ results for each model parameter and $GPP$ with respect to $R_{30d}$ is shown in Fig. 9. $DSS$ results (Fig. 9a) show that for low recharge potential sites q25 and q50, $K_s$ of mud and muddy sand facies were the most sensitive parameters with respect to $R_{30d}$. For high recharge potential sites q75 and q95, $GPP$ was the most sensitive parameter. These findings demonstrate that $K_s$ of fine-texture facies is the dominant driver of recharge potential for low recharge potential sites and the configuration of facies and water table depth is relatively less important. However, for high recharge potential sites,

which presumably have a higher proportion of coarse-texture facies, the configuration of facies and water depth becomes the dominant driver of recharge potential. In general, $DSS$ of all parameters were greater for sites with higher recharge potential than for sites with low recharge potential.

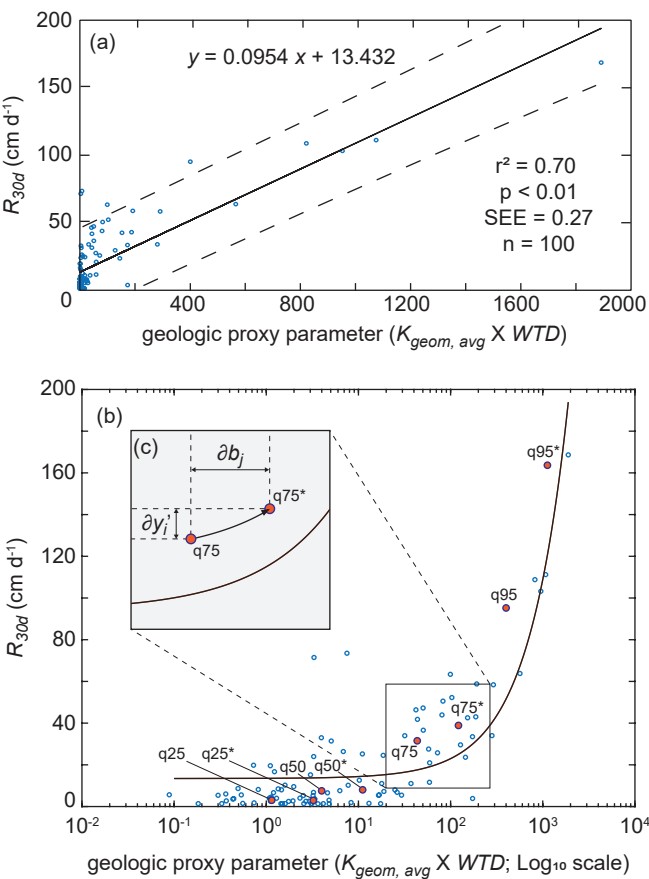

**Figure 7.** (a) Relation between the geologic proxy parameter ($K_{geom} \times WTD$), and the 30-day average recharge rate ($R_{30d}$) for all exploratory simulations, where dashed lines indicate the 95% confidence interval. (b) The relation is shown with $K_{geom} \times WTD$ on a Log$_{10}$ scale, where red circles indicate the original and perturbed sites at which dimensionless scaled sensitivity ($DSS$) was estimated. (c) The inset illustrates the procedure for estimating the perturbed site (e.g., q75*) from the original site (e.g., q75) for $DSS$, using the regression relation, where $\partial y_i'$ is the change in $K_{geom} \times WTD$ and $\partial b_j$ is the estimated corresponding change in $R_{30d}$.

$CSS$ results for $R_{30d}$ (Fig 9b) show that, in general, $GPP$ was the most sensitive parameter for $R_{30d}$ when aggregated across all 4 sites. In general, $K_s$ and $\phi$ were also sensitive with respect to $R_{30d}$. It is unsurprising that $/phi$ is sensitive to $R_{30d}$

because specific yield ($S_y$), which is not explicitly parameterized in ParFlow, is closely related to $\phi$. Moreover, Maples et al. (2019) showed that the majority of recharge volume in this alluvial system is accommodated by filling unsaturated-zone pore volume, which is controlled primarily by $S_y$, and by association in this model, by $\phi$. Empirical fitting parameters describing unsaturated-zone texture and soil water retention, $\alpha$, $n$, and $S_{res}$, were relatively insensitive, especially for sites q75 and q95. This suggests that while unsaturated pore volume is important for recharge, the unsaturated flow processes are not particularly

important, at least when considering infiltration of ponded water, which typically allows for rapid wetting-front advancement

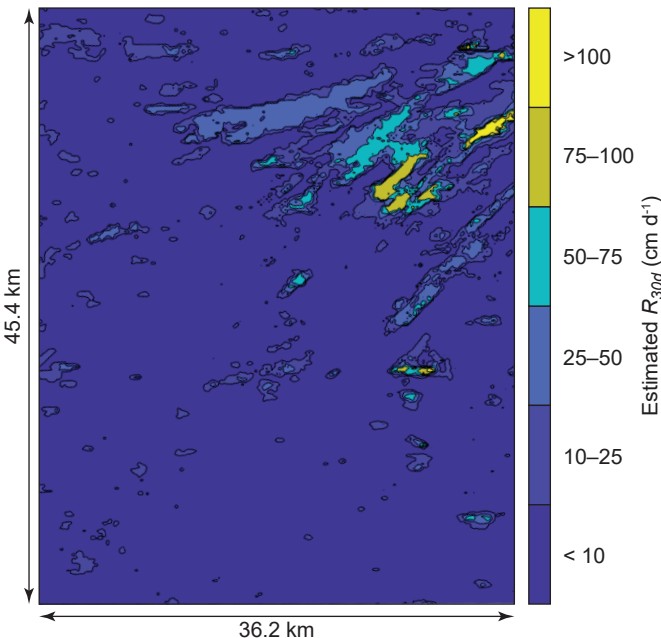

**Figure 8.** Domain-wide estimated distribution of 30-day average recharge rate $R_{30d}$.

through the unsaturated zone, especially for high recharge potential sites. Results suggest that saturated storage properties, i.e., $S_s$, were also relatively unimportant. This likely because most recharge volume is accommodated by filling unsaturated pore volume, and is thus more dependent on $\phi$ (and $S_y$) than on $S_s$.

Normalized $DSS$ for sites q25, q50, q75, and q95, and normalized $CSS$ for all sites are shown in Fig. 10. $DSS$ were scaled

to the range [0,1] (i.e., normalized) for each group of parameters for a given site and a given model output. For example, all $DSS$ values at q25 for $R_{30d}$ were normalized to the maximum value of $DSS$ for that group of parameters. $CSS$ were similarly scaled for each group of parameters for given model output. Because $DSS$ and $CSS$ values are influenced by the units of each model output, normalization allows for comparison of their relative magnitudes between model outputs. Results show similar sensitivity importance for each model output, wherein $K_s$ and $\phi$ are generally the most sensitive parameters, while $S_s$, $\alpha$, $n$,

and $S_{res}$ are all relatively unimportant. $DSS$ and $CSS$ of $GPP$ were not evaluated for $P_{30d}$ and $V_{\text{fines, 90d}}$ because regression relations between site characteristics and these outputs were generally poor compared to those for $R_{30d}$, as noted in section 3.3.

$DSS$ and $CSS$ results for $GPP$ demonstrate the novel usage of empirical regression relations in a local sensitivity analysis framework. By perturbing $GPP$ in this way, constancy of other parameters can be maintained in a way that would be otherwise difficult if $GPP$ was perturbed by changing the location of the recharge site. Performing local sensitivity analyses at multiple

sites spanning a range of recharge potential allowed for comparison of $DSS$ sensitivities across sites and highlights differences of parameter sensitivities for low- and high-recharge potential sites. Our findings demonstrate that (1) facies permeability and unsaturated-zone storage properties are important factors for recharge potential, and (2) the configuration of subsurface

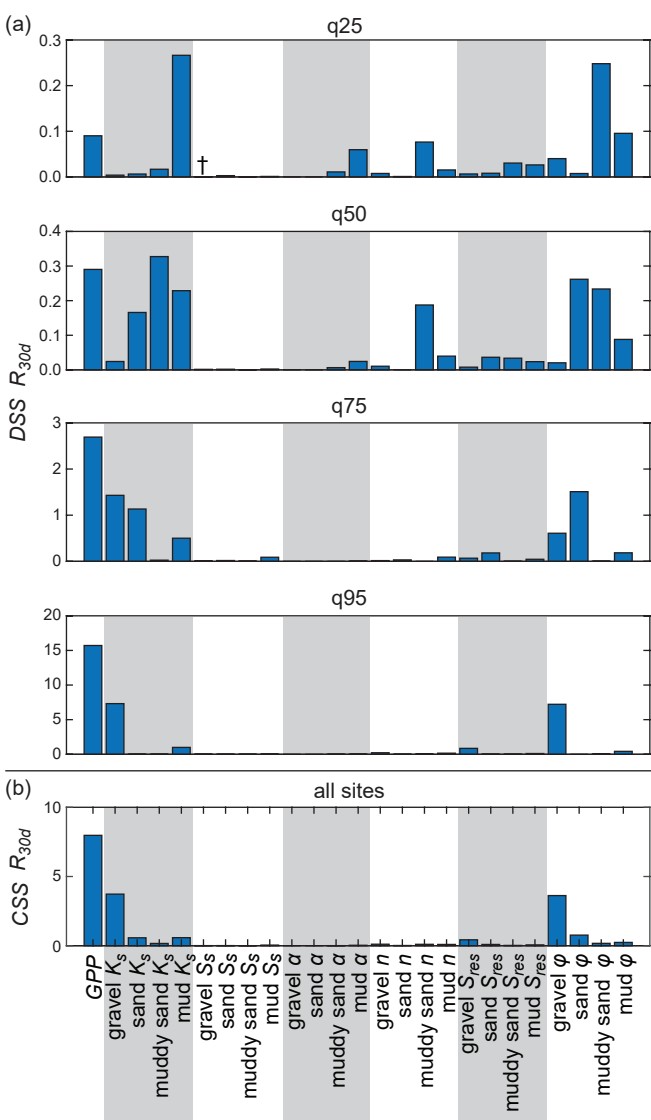

**Figure 9.** (a) Dimensionless scaled sensitivities ($DSS$) evaluated for each model parameter and model output $R_{30d}$ at sites q25, q50, q75, and q95, and (b) composite scaled sensitivities ($CSS$) evaluated for each parameter and model output at all sites. $DSS$ and $CSS$ of parameters were scaled to the range [0,1] (i.e., normalized). † $DSS$ and $CSS$ values below 0.001 are not shown.

geology and water table depth is particularly important for the total recharge volume that can be accommodated at a particular site, especially for high recharge potential sites.

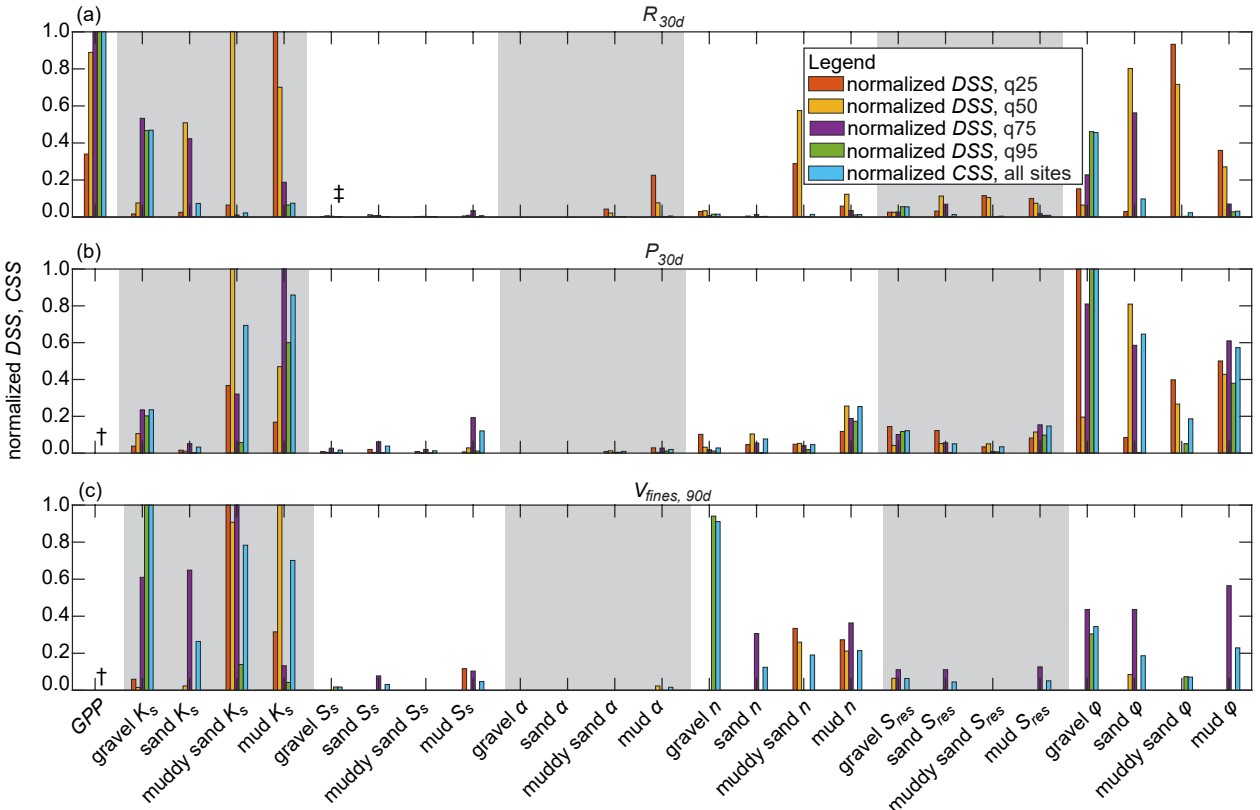

**Figure 10.** Normalized dimensionless scaled sensitivities ($DSS$) evaluated for each model parameter and model outputs (a) $R_{30d}$, (b) $P_{30d}$, and (c) $V_{fines, 90d}$ at sites q25, q50, q75, and q95, and normalized composite scaled sensitivities ($CSS$) evaluated for each parameter and model output at all sites. $DSS$ and $CSS$ of parameters were scaled to the range [0,1] (i.e., normalized). †$DSS$ and $CSS$ of $GPP$ were not evaluated for $P_{30d}$ and $V_{fines, 30d}$. ‡ $DSS$ and $CSS$ values below 0.01 are not shown. (*double-column width*)

### 3.4.2 Global Sensitivity Analyses

Results from global sensitivity analyses are shown in Fig. 11. Morris $\mu*$ values indicate that $GPP$ is the most sensitive parameter when compared with $K_{sat}$ of coarse- and fine-texture facies. These results are consistent with findings from local sensitivity analyses which also showed that $GPP$ was the most important parameter with respect to $R_{30d}$. Unlike $DSS$ and $CSS$ results, which compared $GPP$ against model parameters for each facies, Morris analysis combined $K_s$ parameters for coarse- and fine-texture facies which, in turn, increased the influence of those parameters on $R_{30d}$ relative to $GPP$. Even so, results indicate that $GPP$ is the most important parameter with respect to $R_{30d}$. These results further highlight the importance of the configuration of subsurface geology and water table depth for groundwater recharge potential.

Morris results demonstrate a novel incorporation of $GPP$ within a global sensitivity analysis framework, and was unique as compared to incorporation of $GPP$ in local sensitivity analyses described in section 3.4.1. Unlike the local methods, which




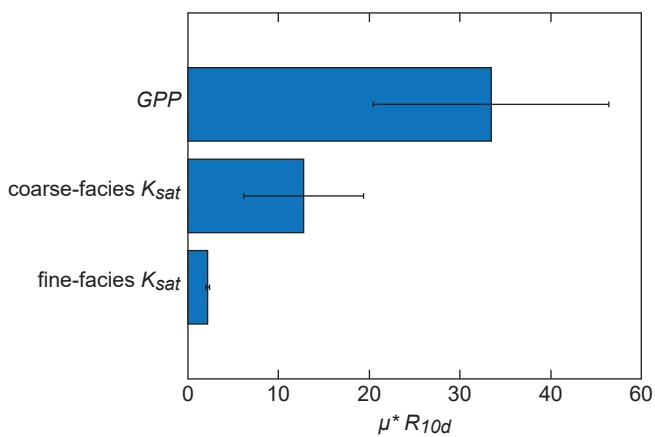

**Figure 11.** Morris $\mu*$ value of $R_{10d}$ for $GPP$ and $K_{sat}$ of coarse-texture (gravel and sand) and fine-texture (muddy sand and mud) facies, where bars represent each $\mu*$ estimate, and whiskers represent the respective 95% confidence interval of the estimate.

used an empirical relation to incorporate an estimate of $GPP$ sensitivity, the method used for the Morris approach directly varied $GPP$ within the parameter space by moving the recharge site to the location with the requisite $GPP$ parameter value. Unlike the local approaches, which required constancy among all other parameters as each parameter is perturbed and thus required usage of an empirical relation to perturb $GPP$, the Morris approach varies all parameters globally, which allowed for $GPP$ to be included explicitly within the approach. Consistency of the results of the local and global approaches despite these
methodological differences for incorporating $GPP$ highlights the robustness of these findings.

## 4 Discussion

Sensitivity analyses are a fundamental diagnostic tool to provide insight into the relative importance of the parameterization of aquifer properties among other inputs to complex hydrologic models (Saltelli et al., 2004). Sensitivity analyses can be broadly categorized as local or global methods, where local methods provide sensitivity evaluation at a single location in the
parameter space (Hill and Tiedeman, 2007), while global approaches explore sensitivities throughout a multi-dimensional parameter space (Saltelli et al., 2008). Many studies have shown the diagnostic utility of local approaches (e.g., Foglia et al., 2009); however, local approaches are generally less robust than global approaches, especially for non-linear models (Saltelli et al., 2008). On the other hand, global methods are typically orders of magnitude more computationally expensive than local approaches. Many studies have evaluated sensitivity of diffuse recharge in hydrologic and landscape models (e.g., Hartmann
et al., 2017; McCallum et al., 2010). Other studies have evaluated sensitivities related to subsurface heterogeneity and permeability upscaling in variably-saturated flow models (e.g., Gilbert et al., 2016; Foster and Maxwell, 2019; Srivastava et al., 2014) and on MAR specifically (e.g., Rahman et al., 2013; Heilweil et al., 2015), but to our knowledge, this is the first study to use a 3-dimensional variably-saturated water flow code with a detailed representation of geologic heterogeneity to evaluate





the importance of hydraulic properties and geologic configuration on MAR dynamics with a combination of local and global
sensitivity analyses.

Our approach includes novel incorporation of geologic architecture as a geologic proxy parameter (i.e., $GPP$). Of the many
approaches to develop a $GPP$ of recharge potential from descriptions of subsurface geologic and hydrologic characteristics,
our results show that a $GPP$ which combines metrics related to upscaled vertical $K_s$ and unsaturated zone thickness was most
diagnostic of recharge potential. In addition, results from local and global sensitivity analyses indicate that this $GPP$ is equally
or more important as characterizing the hydraulic properties of any particular facies for recharge potential. Consistency among
results for both local and global approaches shows that these findings are reasonably robust and highlights the importance of
accurately characterizing the subsurface configuration of coarse-texture facies in clastic sedimentary aquifer systems. While a
$GPP$ was shown to be the most important parameter with both approaches, we also show that parameters related to unsaturated-
zone storage and facies permeability (i.e., $\phi$ and $K_s$, respectively) were also important for MAR. In contrast, we show that
parameters related to unsaturated-zone geologic texture and soil water retention, along with saturated-zone storage properties
(i.e., $\alpha$, $n$, $S_{res}$, and $S_s$) were relatively unimportant. We speculate that these parameters are relatively unimportant because our
simulations typically showed that surface ponding initiated rapid downward wetting-front advancement through the unsaturated
zone, quickly developing fully saturated conditions from land surface to the water table. In systems dominated by diffuse
recharge, these parameters may be more sensitive.

Findings presented here for a semi-confined alluvial aquifer system show large spatial variability of recharge rates that are
dependent primarily on subsurface geologic configuration. we show that select locations in the domain area are capable of
accommodating orders-of-magnitude greater recharge benefit than would be possible over the rest of the landscape. These
findings are consistent with previous studies that indicate that favorable site characteristics, including connect networks of
coarse-texture IVF, are present in the American-Cosumnes River area of the Central Valley, California (Meirovitz, 2010;
Maples et al., 2019), but likely occur over a small fraction of the domain area. Other studies have shown that IVF deposits
occur elsewhere in California's Central Valley (e.g., Weissmann et al., 2005) and in other major river fans that drain high-
elevation, glacially-influenced catchments (e.g., Pierce and Scott, 1983). Identifying sites that can accommodate large MAR
volumes during short windows is especially valuable in places like California, where excess surface water available for recharge
typically occurs from a few precipitation events (Dettinger et al., 2011) during short (< 10 day) windows (Kocis and Dahlke,
465 2017).

Our results show that cursory investigations of soil or surficial geology are likely insufficient to adequately characterize
MAR favorability. Instead, our findings indicate that more thorough investigations of subsurface geologic architecture and
aquifer configuration are needed to accurately characterize MAR feasibility. We found that metrics that consider the geologic
configuration of facies and provide some measure of upscaled vertical $K_s$ are the best predictors of recharge feasibility. We
show that connectivity metrics that determine whether coarse-texture facies interconnect from land surface to the saturated
zone are also helpful, but not fully diagnostic of recharge potential. Interestingly, our results show that metrics describing
unsaturated-zone thickness, fraction of coarse-texture facies at land surface, and fraction of coarse-texture unsaturated-zone
facies are insufficient when each is considered alone. This finding has important implications because several GIS-derived





metrics of recharge potential describing recharge suitability of surficial soils have been developed for California and elsewhere

(O'Geen et al., 2015; Adham et al., 2010; Ghayoumian et al., 2007). We consider these products as valuable, albeit incomplete metrics that are likely complemented by more detailed investigations of deeper subsurface geologic architecture.

Importantly, no single $GPP$ described herein was a fully diagnostic metric of recharge potential at all sites. This result is not surprising given the complexity of geologic architecture and variability of aquifer configuration sampled across sites in the domain, which are challenging to fully captured with a single metric. For example, all site characteristics described

here were developed only for those model cells that are vertically-coincident with each site footprint, and do not account for possible preferential pathways in adjacent cells outside of the immediate site footprint. We acknowledge that further research into this phenomena could provide additional insight into developing site-specific $GPP$, but is outside the immediate scope of this work. We also acknowledge some limitations of our sensitivity analyses. For example, reliance on imperfect empirical regression relations to include measures of geologic configuration in local methods likely introduced uncertainty to $DSS$ and

$CSS$ estimates for this parameter. In addition, inclusion of all model parameters describing facies hydraulic properties in the Morris approach would have been valuable, but was infeasible given the computational resources for the simulations required. In addition, parameter-range uncertainty contributes some uncertainty to rankings of parameter importance.

Our simulations also do not consider some subsurface geologic conditions that influence MAR. Clastic sedimentary aquifer systems are typically replenished naturally over longer timescales (Taylor et al., 2013) because even productive aquifer systems

are commonly composed mostly of fine-texture sediments (e.g., Fogg, 1986; Fogg et al., 2000), that form nearly ubiquitous, multiple confining layers that inhibit direct recharge of the interconnected sand and gravel body networks that comprise the aquifer system. The presence of laterally-continuous aquitard facies have been well documented portions of the southern Central Valley (Phillips and Belitz, 1991; Faunt et al., 2009), and in other unconsolidated alluvial aquifer systems in California (e.g., Fisher, 1964). While not present within the domain area, these features have been shown to uniformly impede recharge

to confined aquifer systems where they are present. In addition, we do not consider some surface conditions that affect real-world MAR, like topographic site limitations, evaporative losses, and clogging effects (Bouwer, 2002). We emphasize that this study is not a thorough site investigation of the American-Cosumnes area. The TPROGS approach is inherently stochastic and conditioning data to inform the model are sparse in places (Maples et al., 2019). In addition, the single TPROGS realization used for our simulations provides only a single representation of possible facies distributions within the domain. Our find-

ings are presented as a proof-of-concept to explore the importance of geologic heterogeneity on MAR in a hypothetical but physically-realistic domain.

Our findings have important implications for assessing MAR feasibility and for understanding MAR processes in clastic alluvial aquifer systems in California and globally, where accelerating groundwater overdraft and increasing water scarcity are observed (Scanlon et al., 2012; Famiglietti et al., 2011; Wada et al., 2011). Our results highlight the importance of identifying

and cataloging locations with favorable geology for recharge, especially in light of recently-passed groundwater management legislation in California that mandates limiting both the "chronic lowering of groundwater levels" and "significant and unreasonable reductions in groundwater storage" (Kiparsky et al., 2016). While studies have shown that implementation of MAR can lead to more sustainable groundwater management (e.g., Niswonger et al., 2017), widespread adoption of of MAR is





still hampered by a number of challenges, including institutional barriers to water-rights transference and water accounting
uncertainty (Asano, 2016), infrastructure limitations, including land acquisition and water conveyance costs (Gailey, 2018),
and water quality considerations (Hartog and Stuyfzand, 2017). Our approach, which combines a detailed representation of
subsurface geology with physically-realistic water flow physics in a sensitivity analysis framework, can (1) help guide site
investigations and data collection methods for proposed MAR projects, and (2) improve representation of recharge processes
in management-focused, typically coarse-resolution groundwater models.

## 5   Conclusions

This research explores the sensitivity of hydraulic properties and subsurface geologic architecture on MAR processes with the
variably-saturated water flow code, ParFlow, in a highly heterogeneous geologic domain that reflects the complex, unconsol-
idated alluvial geologic architecture of the northern Central Valley, CA that is consistent with many alluvial aquifer systems.
This work comprises two fundamental components. First, exploratory simulations were performed at 100 randomly-sampled
sites across the domain to to evaluate the correlation between 17 geologic and hydrologic site characteristics and simulated
recharge benefits. Results from the exploratory simulations show that site characteristics representing subsurface geologic con-
figuration by upscaling vertical $K$ can produce good correlations with the average 30-day recharge rate ($R_{30d}$). Regression re-
lations between site-averaged $K_{geom} \times WTD$ and $R_{30d}$ were shown to be the most correlated ($r = 0.70$, $p < 0.01$, $r^2 = 0.70$).
Conversely, site characteristics describing unsaturated-zone thickness ($WTD$), fraction of coarse-texture unsaturated-zone
($UZ_{coarse}$), and fraction of coarse-texture surface facies ($Surf_{coarse}$) alone were all poorly correlated with $R_{30d}$. These re-
sults highlight the value of characterizing subsurface geologic configuration through $K$ upscaling. For subsequent sensitivity
analyses, $K_{geom} \times WTD$ was designated as a geologic proxy parameter, $GPP$, for recharge potential using aforementioned
predictive regression relation.

Subsequent local and global sensitivity analyses were performed for model hydraulic properties and the $GPP$ to evaluate
the relative importance of these parameters on recharge potential. Results from local sensitivity analyses indicated that $GPP$ is
the most sensitive parameter for $R_{30d}$, more so than any parameters describing hydraulic properties of each facies. Sensitivity
analyses also indicated that permeability and unsaturated-zone pore volume (i.e., $K_s$ and $\phi$, respectively) were relatively more
important than other hydraulic properties, including unsaturated-zone geologic texture, soil water retention, and saturated-zone
storage properties (i.e., $\alpha$, $n$, $S_{res}$, and $S_s$) for $R_{30d}$. Results from global sensitivity analyses were consistent with local sensitiv-
ity analyses, indicating that $GPP$ is relatively more important than $K_s$ of coarse- and fine-texture facies for $R_{30d}$. Agreement
of local and global approaches regarding the importance of $GPP$ shows a degree of robustness of these findings. The results
presented here demonstrate the importance of thoroughly characterizing subsurface geologic configuration when considering
recharge feasibility. To our knowledge, this study is the first of its kind to incorporate of a measure of geologic configuration
with a geologic proxy parameter in formal sensitivity analyses. Our approach outlines a novel combination of subsurface site
characterization with simulations of variably-saturated water flow physics within a sensitivity analysis framework to (1) im-



prove understanding the role of geologic heterogeneity on MAR processes and (2) provide insight into potential strategies to characterize subsurface geologic heterogeneity when considering recharge feasibility.





*Author contributions.* Conceptualization: SM, LF, GF, & RM; Analysis: SM; Writing-Original Draft Preparation: SM; Writing-Review & Editing: SM, LF, GF, & RM; Funding Acquisition: SM, LF, GF, & RM

*Competing interests.* The authors declare no conflict of interest. The funders had no role in the design of the study; in the collection, analyses, or interpretation of data; in the writing of the manuscript, or in the decision to publish the results

*Acknowledgements.* We gratefully thank Jon Herman, Mary Hill, Robert Reinecke, Lauren Thatch, Mary Michael Forester, Nick Engdahl, and Thomas Harter for their assistance, along with anonymous reviewers for helpful comments on the manuscript. Support for this research was provided by the National Science Foundation (NSF) Climate Change, Water, and Society (CCWAS) Integrated Graduate Education
and Research Traineeship (IGERT) program at the University of California, Davis and Colorado School of Mines (http://ccwas.ucdavis.edu, DGE-10693333) (SM, GF, RM), the NSF Graduate Research Fellowship (SM), the University of California Water (UC Water) Security and Sustainability Research Initiative (SM, GF, LF). We would like to acknowledge high-performance computing support from Cheyenne (doi:10.5065/D6RX99HX) provided by NCAR's Computational and Information Systems Laboratory, sponsored by NSF. All data used in the analysis can be made available by SM (srmap@ucdavis.edu).





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



**Table 1.** Textural classification of hydrofacies designations (Fleckenstein et al., 2004)

| Hydrofacies | Geologic Interpretation | Texture |
|---|---|---|
| Gravel | Channel deposits | Gravel and coarse sand |
| Sand | Near channel/levee | Sand (fine to coarse) |
| Muddy Sand | Proximal floodplain | Silty and clayey sand, sandy clay, and silt |
| Mud | Floodplain | Clay, silty clay, shale |





**Table 2.** Hydrofacies Hydraulic Properties (Liu, 2014)

| Facies Designation | $K_s$ (m d$^{-1}$) | $S_s$ (m$^{-1}$) | $\phi$ (unitless) | $\alpha$ (unitless) | $n$ (unitless) | $S_{res}$ (unitless) |
|---|---|---|---|---|---|---|
| Gravel | 67.5 | 4.00E-05 | 0.35 | 3.55 | 3.16 | 0.1 |
| Sand | 41.2 | 8.00E-05 | 0.35 | 3.55 | 3.16 | 0.1 |
| Muddy Sand | 0.2 | 1.00E-04 | 0.40 | 2.69 | 2.00 | 0.1 |
| Mud | 0.0017 | 1.00E-03 | 0.45 | 1.62 | 2.00 | 0.2 |
| Deep Aquifer | 45.0 | 4.80E-04 | 0.35 | 3.55 | 3.16 | 0.1 |





**Table 3.** Descriptions of site characteristics (variables) and model outputs (predictors)

| Observation type | Name | units | Description |
|---|---|---|---|
| site characteristics (variables) | $K_{\text{arith, max}}$ | m d$^{-1}$ | Maximum arithmetic mean of vertical $K_{\text{s}}$ for cells above the initial water table for each site |
| | $K_{\text{geom, max}}$ | m d$^{-1}$ | Maximum geometric mean of vertical $K_{\text{s}}$ for cells above the initial water table for each site |
| | $K_{\text{harm, max}}$ | m d$^{-1}$ | Maximum harmonic mean of vertical $K_{\text{s}}$ for cells above the initial water table for each site |
| | $K_{\text{arith, avg}}$ | m d$^{-1}$ | Average arithmetic mean of vertical $K_{\text{s}}$ for cells above the initial water table for each site |
| | $K_{\text{geom, avg}}$ | m d$^{-1}$ | Average geometric mean of vertical $K_{\text{s}}$ for cells above the initial water table for each site |
| | $K_{\text{harm, avg}}$ | m d$^{-1}$ | Average harmonic mean of vertical $K_{\text{s}}$ for cells above the initial water table for each site |
| | $UZ_{\text{coarse}}$ | unitless | Proportion of cells above the initial water table that are coarse-texture facies (gravel and sand) for each site |
| | $Surf_{\text{coarse}}$ | unitless | Proportion of surface cells that are coarse-texture facies (gravel and sand) for each site |
| | $WTD$ | m | Average initial water table depth for each site |
| model outputs (predictors) | $R_{\text{10d}}$ | cm d$^{-1}$ | Effective recharge rate (0–10 day average) |
| | $R_{\text{30d}}$ | cm d$^{-1}$ | Effective recharge rate (0–30 day average) |
| | $P_{\text{30d}}$ | m$^3$ | Volumetric extent of pressure perturbation after 30 days (10-cm threshold) |
| | $V_{\text{90d, fines}}$ | unitless | Proportion of total recharge volume accommodated by fine-texture facies after 90 days |



**Table 4.** Parameter ranges, along with baseline and $DSS$ perturbed parameter values for hydraulic properties and the geologic proxy parameter ($GPP$)

| Parameter | Facies Designation | Parameter Range† | Baseline Parameter‡ | $DSS$ Perturbed Parameter |
|---|---|---|---|---|
| $K_s$ (m d$^{-1}$) | Gravel | $14.42 - 144.2$ | 67.5 | 80.48 |
| | Sand | $5.4 - 54.0$ | 41.2 | 46.06 |
| | Muddy Sand | $0.089 - 0.89$ | 0.20 | 0.28 |
| | Mud | $0.0023 - 0.023$ | 0.0017 | 0.0038 |
| $S_s$ (m$^{-1}$) | Gravel | $1.0\times10^{-6} - 1.0\times10^{-4}$ | $4.0\times10^{-5}$ | $4.9\times10^{-5}$ |
| | Sand | $1.0\times10^{-6} - 1.0\times10^{-4}$ | $8.0\times10^{-5}$ | $8.9\times10^{-5}$ |
| | Muddy Sand | $0.0001 - 0.001$ | 0.0001 | 0.0002 |
| | Mud | $0.0001 - 0.001$ | 0.0010 | 0.0011 |
| $\alpha$ (unitless) | Gravel | $3.55 - 3.55$ | 3.55 | 3.55 |
| | Sand | $3.55 - 3.55$ | 3.55 | 3.55 |
| | Muddy Sand | $2.69 - 3.55$ | 2.69 | 2.78 |
| | Mud | $0.35 - 0.45$ | 1.62 | 1.81 |
| $n$ (unitless) | Gravel | $2.00 - 3.16$ | 3.16 | 3.044 |
| | Sand | $1.89 - 3.16$ | 3.16 | 3.033 |
| | Muddy Sand | $1.44 - 2.00$ | 2.00 | 1.94 |
| | Mud | $1.32 - 2.00$ | 2.00 | 1.93 |
| $S_{res}$ (unitless) | Gravel | $0.10 - 0.14$ | 0.10 | 0.104 |
| | Sand | $0.10 - 0.14$ | 0.10 | 0.104 |
| | Muddy Sand | $0.10 - 0.25$ | 0.10 | 0.12 |
| | Mud | $0.16 - 0.23$ | 0.20 | 0.21 |
| $\phi$ (unitless) | Gravel | $0.25 - 0.35$ | 0.35 | 0.34 |
| | Sand | $0.25 - 0.35$ | 0.35 | 0.34 |
| | Muddy Sand | $0.35 - 0.45$ | 0.40 | 0.39 |
| | Mud | $0.35 - 0.45$ | 0.45 | 0.44 |
| $GPP$ (m$^2$ d$^{-1}$) | - | $0.08 - 1891.6$ | - | - |

† Ranges of hydraulic properties for each facies category were derived from literature values for the Central Valley, California, and for similar alluvial systems (Anderson et al., 2015; Botros et al., 2009; Fleckenstein et al., 2004; Frei et al., 2009; Maserjian, 1993; Niswonger and Fogg, 2008; Sager, 2012).

‡ Baseline hydraulic properties were calibrated manually by Liu (2014)