# Peer review of "Sensitivity of Hydrologic and Geologic Parameters on Recharge Processes in a Highly-Heterogeneous, Semi-Confined Aquifer System"

_Hydrology and Earth System Sciences, 2019_

## Referee Comment (RC1) · Anonymous Referee #1 · 23 Sep 2019

This manuscript presents an assessment study of the hydrologic and geologic impact on managed aquifer recharge processes. At 100 randomly sampled sites across the model domain the correlation between 17 hydro(geo)logical site characteristics/parameters and simulated recharge "benefits" is evaluated. Overall, upscaled vertical K multiplied with "Water Table Depth" (WTD) produce a good correlation with recharge rates. This proxy parameter (GPP – Kgeom * WTD) are most correlated with recharge rates, validated by local and global sensitivity analysis. Moreover, the analyses also indicate that permeability and unsaturated zone pore volume (porosity used as an indicator for Sy) were relatively more important than other hydraulic parameters.

The study presented is comprehensive, thorough, well-organized, and clear. The conclusions are informative and I do not see any over-statement in the conclusions drawn. I thus think the manuscript should be considered for publication in HESS, although I would suggest that a minor revision is needed to clarify some parts of the manuscript.

Maybe the most critical point I see is how to transfer the obtain important information for MAR (interconnected coarse-texture facies paired with water table depth information are crucial for finding suitable recharge sites) to any other field where this information is difficult to acquire. I see your point that GIS-derived indices of recharge suitability rely solely on soil and surface geology to determine geologic suitability for recharge (e.g. Line 354) and the integrated values (up-scaled K + WTD) are more useful. But the question is how we could get the required information without knowing the subsurface in every detail in the whole model domain/study area. Certainly, in your (semi-)synthetic approach we know the parameterization (by the way it is just one field/realization and there remain uncertainties about the distribution, however, for this study and target it is ok I believe but should be note more clearly in the discussion) but how could we use your guidelines where it is not known. I suggest discussing that more to strengthen the manuscript and impact of this interesting study. Comprehensive field tests could be probably the best to better understand the problems in general. However, they are time-consuming and only a limited number of sites are available to accommodate the tests. The numerical analysis, on the other hand, allows us to explore and assess multiple sites relatively easily, yet the validity needs to be carefully checked. So, how can your useful guideline to be considered by practitioners?

Another point is related to the WTD. Correct me if I am wrong but my impression based on your manuscript (for instance Fig. 8) is that Kgem * WTD is very useful where the WTD is deep (so large unsaturated zone and thus more storage). Can you split your analysis/results further to see if the depth to the water table matters or not?

Further minor comments: Line 94: What kind of geological analysis?

Section 2.3.3 Model Spin up and Calibration: All sentence related to boundary conditions should be moved to section 2.3.2 Boundary conditions. (line 134-141)

Table 2: Are these parameters the calibrated values?

Section 2.4.1 Why do you select the sites randomly? I would assume that MAR will be pretty much every time in more coarse sediments. I think, a useful comparison would be to choose the sites based on the surface information (as you mentioned as the "classical" GIS-approach) and compare the results with results from some randomly chosen sites. You might find additional arguments to criticize the "classical" workflow. I think that would be "just" another post-processing step and no demanding model runs are required.

Line 201: for all 100 sites

Line 230: 6 hydraulic properties and not 8!

Line 258: Where are the four representative sites! You could add these sites to figure 3a.

Line 286: yes, they are important, but it is not demonstrated here. The results section just comes a few pages later. Please reformulate.

Line 3357: Yes, but again how to get this information for a larger study site.

Figure 7: Change the 95% confidence lines to dashed lines or change the figure caption.

Figure 10: Why is the gravel n so important for V_fines?
* * *

---

## Referee Comment (RC2) · Anonymous Referee #2 · 25 Oct 2019

**General comments**

The manuscript "Sensitivity of Hydrologic and Geologic Parameters on Recharge Processes in a Highly-Heterogeneous, Semi-Confined Aquifer System" describes an interesting study on local and global sensitivity analysis in the framework of Managed Aquifer recharge, using a realistic case study. Overall, the manuscript is well written and the results are illustrated in a clear manner. Although the research work heavily relies for the creation of the geological model and the setting up of a flow model and MAR on two previous works, the additional research performed in this study and the new findings justify a new publication. I only have a couple of minor suggestions and

some technical details.

**Specific comments**

**Control volume and connectivity metric (lines 179-188; 331-333)** Please double check the definition of the control volume and the need for a 6-points connectivity metric: if the control volume is defined as "encompassing vertically-coincident cells" (line 179), then there is probably no need to require a 6-points connectivity metric. For example, with a 6-points connectivity, you can have 2 very horizontally extended layers of a conductive material, separated by a rather impermeable aquitard; if only one cell of the aquitard is conductive, then the 6-points connectivity guarantees connection. Maybe I missed the definition of the control volume. Is it defined by only one cell in the horizontal directions?

**Linearity (197-200)** As your aquifer is not confined, maybe the fact of separating the contribution of each recharge/no-recharge scenario would not work properly as in the case of a linear problem. Please comment on this.

$r$ **sign** In general, for a negative correlation a negative $r$ is used (line 315, but also the corresponding figures).

**Figures**

**Fig.1 and Fig.3** Please report the original publication source of the figure.

**Fig.8** Do you also have a map of IVF? It would be nice to see it on the side of the $R_{30d}$ (see also line 370).

**HESSD**

**Technical corrections**

**line 63** 1640 m$^2$

**Parenthesis** Double check journal guidelines for parentheses (i.e. lines 67, 76-77, 87, 91)

**Units repetition** It would be more correct to report units close to each number, for example $1 \times 2 \times 3\,\mathrm{m}$ should be something like $1\,\mathrm{m} \times 2\,\mathrm{m} \times 3\,\mathrm{m}$ (see lines 99, 101). This is also valid when a list of numbers (with unit) is reported. See for example line 308, 309, 320.

**Subscript fonts** In general, subscripts that are not index should not be in italic font (i.e., $S_s$ should be $S_\mathrm{s}$ instead) (see line 110 and other locations in the text)

**lines 123-124** Check "0 m amsl".

**UZ (line 141)** Please introduce this acronym.

**line 165-166** $S_s$ or $S_S$?

$V_\mathrm{fines,90d}$ Double check the consistency of this symbol within the documents (see for example Fig.5).

**line 479** "to *be* fully..."?

**line 538** "to incorporate a measure"

---

## Author Comment (AC2) · 25 Nov 2019

GENERAL COMMENTS

COMMENT: "The manuscript "Sensitivity of Hydrologic and Geologic Parameters on Recharge Processes in a Highly-Heterogeneous, Semi-Confined Aquifer System" describes an interesting study on local and global sensitivity analysis in the framework of Managed Aquifer recharge, using a realistic case study. Overall, the manuscript is well written and the results are illustrated in a clear manner. Although the research work heavily relies for the creation of the geological model and the setting up of a flow model and MAR on two previous works, the additional research performed in this study

and the new findings justify a new publication. I only have a couple of minor suggestions and some technical details." RESPONSE: Thank you for the comments. We have addressed your comments, and have provided responses to each specific comment below.

SPECIFIC COMMENTS:

COMMENT: "Control volume and connectivity metric (lines 179-188; 331-333) Please double check the definition of the control volume and the need for a 6-points connectivity metric: if the control volume is defined as "encompassing vertically-coincident cells" (line 179), then there is probably no need to require a 6-points connectivity metric. For example, with a 6-points connectivity, you can have 2 very horizontally extended layers of a conductive material, separated by a rather impermeable aquitard; if only one cell of the aquitard is conductive, then the 6-points connectivity guarantees connection. Maybe I missed the definition of the control volume. Is it defined by only one cell in the horizontal directions?" RESPONSE: We agree that the definition of the control volume was unclear, and have made substantial changes to section 2.4.2 to re-frame and add detail to how site characteristics are presented. For example, we have added the sentence (lines 187-189): "Percolation was evaluated for a control volume encompassing all cells from the land surface to the initial water table depth (i.e., unsaturated-zone cells) at the 25 x,y cell locations encompassing each site." Because each control volume incorporates both vertically- and horizontally-connected cells, the 6-connectivity metric is necessary to evaluate percolation. We believe that the clarification regarding the definition of the control volume will make this clear to the reader.

COMMENT: "Linearity (197-200) As your aquifer is not confined, maybe the fact of separating the contribution of each recharge/no-recharge scenario would not work properly as in the case of a linear problem. Please comment on this." RESPONSE: Thank you for pointing out some of the limitations associated with the differencing approach we use to post-process the results. We have added several sentences to the Discussion section to acknowledge the limitations of this approach for non-linear models, and also

noted that we did not encounter spurious recharge stresses or unrealistic model noise when using this approach with our model. (lines 517-521)

COMMENT: "r sign In general, for a negative correlation a negative r is used (line 315, but also the corresponding figures)." RESPONSE: Thank you for catching this mistake. We added text to point out that R_10d, R_30d, and P_30d were positively correlated with all simulated outputs, but V_fines, 90d was generally negatively correlated with simulated outputs. We made changes throughout the text to report negative correlations appropriately, and have modified Fig. 5 to make clear that correlations are reported as the absolute value.

FIGURES

COMMENT: "Fig.1 and Fig.3 Please report the original publication source of the figure." RESPONSE: We have reported the original source of Figure 1 (Maples et al., 2019), but Figure 3 is unique to this publication, and thus does not have a publication source, so the Figure 3 caption was left as-is.

COMMENT: "Fig.8 Do you also have a map of IVF? It would be nice to see it on the side of the R30d (see also line 370)." RESPONSE: We chose not to overlay a map of IVF on our stochastic geologic model because our findings are presented as a proof-of-concept of a hypothetical, but physically-realistic domain (see lines 515-517). Instead, we rely on citations of the relevant studies that have identified IVF in this region and encourage the reader to seek those publications for additional information.

TECHNICAL CORRECTIONS

COMMENT: "line 63 :1640 m2" RESPONSE: Thank you for catching this mistake. We have changed "1640mˆ2" to "1640 mˆ2"

COMMENT: "Parenthesis: Double check journal guidelines for parentheses (i.e. lines 67, 76-77, 87, 91)" RESPONSE: We have checked the manuscript preparation guidelines for HESS and did not find specific guidance for these instances of parentheses.

[Figure]

We will defer to the associate editor and copy editor to provide guidance on these instances.

COMMENT: "Units repetition. It would be more correct to report units close to each number, for example 1 × 2 × 3 m should be something like 1 m × 2 m × 3 m (see lines 99, 101). This is also valid when a list of numbers (with unit) is reported. See for example line 308, 309, 320." RESPONSE: Thank you for catching this mistake. We have made changes here and elsewhere throughout the manuscript to correct unit repetition issues.

COMMENT: "Subscript fonts In general, subscripts that are not index should not be in italic font (i.e., Ss should be Ss instead) (see line 110 and other locations in the text)" RESPONSE: Thank you for catching this mistake. We have made changes here and elsewhere throughout the manuscript to correct italicization mistakes in subscripts.

COMMENT: "lines 123-124 Check '0 m amsl'." RESPONSE: We are leaving the acronym as-is because above mean sea level (amsl) is introduced previously in Section 2.1 (line 77)

COMMENT: "UZ (line 141) Please introduce this acronym." RESPONSE: Thank you for catching this mistake. We have changed "near-surface UZ cells" to "near-surface unsaturated-zone (UZ) cells"

COMMENT: "line 165-166 Ss or SS?" RESPONSE: Thank you for catching this mistake. We have changed "K_S" to "K_s"

COMMENT: "Vfines,90d Double check the consistency of this symbol within the documents (see for example Fig.5)." RESPONSE: Thank you for catching this mistake. We have fixed the figure accordingly, and have double-checked the consistency of its usage throughout the text.

COMMENT: "line 479 "to be fully..."? " RESPONSE: Thank you for catching this mistake. We have changed "challenging to fully captured" with "challenging to fully cap-
ture."

COMMENT: "line 538 "to incorporate a measure"" RESPONSE: Thank you for catching this mistake. We have incorporated the edit.

———————————————

Correlation (absolute value of Pearson's *r*)

[Figure]

**Fig. 1.**

---

## Author Comment (AC3) · 25 Nov 2019

[revised manuscript text omitted]
 as well as the 4 sites chosen for sensitivity analyses (q95, q75, q50, and q25; solid violet squares), along with (b) arithmetic, (c) geometric, and (d) harmonic mean of vertical $K_{sat}$ for unsaturated zone (UZ) facies (i.e., $K_{geom}$, $K_{arith}$, $K_{harm}$, respectively). The second row shows (e) the coarse-texture (gravel and sand) fraction of UZ facies ($UZ_{coarse}$), (f) simulated initial depth-to-water ($WTD$), and (g) $K_{geom}$ multiplied by $WTD$ ($K_{geom} \times WTD$). Locations >5 km from the lateral domain boundaries were excluded from the potential sites to avoid interference with boundary conditions. *(double-column width)*

41,087 x,y cell locations across the domain, where the arithmetic and harmonic mean ($K_{arith}$ and $K_{harm}$) are the upper and lower bounds, respectively, and the geometric mean ($K_{geom}$) is an intermediate value. $K_{arith}$ and $K_{harm}$ are typically used to approximate groundwater flow parallel and perpendicular to layering, respectively, in anisotropic systems (Freeze and Cherry, 1979). This concept has been generally been extended to variably-saturated flow  (Mualem, 1984; Yeh et al., 1985a, b; Assouline and Or, 2006). Fogg et al. (2000) showed that vertical groundwater flow in

systems with vertically-connected networks of permeable facies tends toward values between $K_{arith}$ and $K_{geom}$.  For each x,y cell location across the domain, $K_{arith}$, $K_{geom}$, and $K_{harm}$  are given as:

$$K_{arith} = \frac{K_1 + K_2 + ... + K_n}{n} \tag{3}$$

$$K_{geom} = \sqrt[n]{K_1 \times K_2 \times ... \times K_n} \tag{4}$$

$$K_{harm} = \frac{n}{\frac{1}{K_1} + \frac{1}{K_2} + ... + \frac{1}{K_n}} \tag{5}$$

 where $n$ is the number of vertically-coincident cells from the land surface to the initial water table depth (i.e., the unsaturated-zone cells) for each x,y cell location. In addition, the initial unsaturated-zone thickness ($WTD$)  and proportion of coarse-texture sand and gravel unsaturated-zone facies ($UZ_{coarse}$) ~~, and proportion of coarse-texture facies at land surface ($Surf_{coarse}$). Each site characteristic was calculated for a control volume encompassing vertically-coincident cells from the land surface to the initial water table depth for each site. For the purposes of this work, the interface of the deeper aquifer system was designated as the initial water table depth ($WTD$). Additional site characteristics were developed by combining existing site characteristicssite characteristics~~ metrics at all x,y cell locations are shown in Fig. 3.

*TABLE 3 ABOUT HERE.*

Site characteristics were calculated for the 100 1 km$^2$ exploratory sites from these metrics by determining the average and maximum value of $
[revised manuscript text omitted]

[Figure]

**Figure 4.** Box plots of the (a) 30-day average recharge rate, $R_{30d}$ and (b) 30-day pressure perturbation area of influence, $R_{30d}$ for all exploratory simulations ($n = 100$). Additionally, sites were parsed according to whether there was vertical interconnection of coarse-texture facies from land surface to the initial water table depth (i.e., interconnected sites, $n = 23$), or whether sites did not have interconnection of coarse-texture facies (i.e., disconnected sites, $n = 77$).

[Figure]

**Figure 5.** Correlations (absolute value of Pearson's $r$) for all combinations of site characteristics and model outputs. Correlations among site characteristics are bounded by a solid red box, and correlations between site characteristics and model outputs are bounded by a dashed red box.

simulated outputs. Strong correlation ( $|r| > 0.70$) was observed for 6 of 52 pairs of site characteristics and simulated outputs. Strong correlation was also observed among many site characteristics and among the majority of simulated outputs (i.e, collinearity), which can make the choice of an optimal proxy parameter more challenging. Site characteristics that include

360    $K_{\text{harm}}$ were not shown in the correlation matrix because we were not able to improve normality of the distribution these data with a $\text{Log}_{10}$ data transformation; however, additional correlation metrics (Fig. 6) indicate that site characteristics that include $K_{\text{harm}}$ may also be strongly correlated.

**3.2.2   Ranked Correlations**

Additional correlation metrics (Pearson's $r$, Spearman's rho, and Kendall's tau) between $R_{30d}$ and site characteristics were

365   ranked and are shown in Fig. 6. Results show that site characteristics that include some variation of $K_{\text{arith}}$, $K_{\text{geom}}$, or $K_{\text{harm}}$ were, in general, more correlated with $R_{30d}$ than site characteristics that only include $WTD$, $UZ_{\text{coarse}}$, and $Surf_{\text{coarse}}$. $K_{\text{geom}} \times WTD$ was, on average, most correlated with $R_{30d}$.

In general, site characteristics that included $K_{\text{geom}}$ and $K_{\text{harm}}$ were slightly more correlated with $R_{30d}$ than site characteristics that included $K_{\text{arith}}$. We speculate that this behavior is related to the dominantly vertical flow direction of recharge across

[Figure]

**Figure 6.** Ranked correlations (Pearson's $r$, Spearman's Rho, and Kendall's Tau) of site characteristics with 30-day average recharge rate ($R_{30d}$). †Pearson's $r$ was not evaluated for site characteristics where the normality of the distribution could not be improved with a $\text{Log}_{10}$ data transformation.

370 typically horizontal facies configurations. Previous work has shown that $K_{geom}$ and $K_{harm}$ best describe upscaled $K$ for these flow configurations in this domain (Yunjie Liu, personal communication; Fogg, 1986).

  Interestingly, site characteristics that included only $WTD$, $UZ_{coarse}$, and $Surf_{coarse}$ were poorly correlated ($r < 0.20$) with $R_{30d}$. This finding has important implications for determining MAR site suitability because many GIS-derived indices of recharge suitability rely solely on soil and/or surface geology to determine geologic suitability for recharge. These results

375 suggest that even more detailed geologic descriptions that estimate deeper fractions of coarse-texture facies may not fully capture recharge potential. Instead, metrics that include some description of upscaled vertical $K$ appear to be most diagnostic of recharge potential.

**3.3 Recharge Extrapolation**

  The relation between site-averaged $K_{geom} \times DTW$ and $R_{30d}$ was determined to be the best predictor and was used to predict

380 $R_{30d}$ for subsequent sensitivity analyses by treating $K_{geom} \times DTW$ as a $GPP$ (Fig. 7a). The linear regression relation between $K_{geom} \times DTW$ and $R_{30d}$ was highly significant ($p < 0.01$), and correlation coefficients ($r^2$) showed that empirical regression explained 70% of the variation in the data. Linear regression relations for $K_{geom} \times DTW$ and $P_{30d}$ and $V_{fines, 90d}$ were deemed insufficient for prediction ($r^2 < 0.40$) and were not incorporated in sensitivity analyses.

[Figure]

**Figure 7.** (a)  Regression relation between the geologic proxy parameter ($K_{\text{geom}} \times WTD$)  and the 30-day average recharge rate ($R_{30d}$) for all exploratory simulations shown with a solid black line, where dashed lines indicate the upper and lower 95% confidence  intervals. (b) The relation is shown with $K_{\text{geom}} \times WTD$ on a Log$_{10}$ scale, where red circles indicate the original and perturbed sites at which dimensionless scaled sensitivity ($DSS$) was estimated. (c) The inset illustrates the procedure for estimating the perturbed site (e.g., q75*) from the original site (e.g., q75) for $DSS$, using the regression relation, where $\partial y_i'$ is the change in $K_{\text{geom}} \times WTD$ and $\partial b_j$ is the estimated corresponding change in $R_{30d}$.

[revised manuscript text omitted]

‡ Baseline hydraulic properties were calibrated manually by Liu (2014)

---

## Author Response (AR1)

Notes to Reviewer 1: Author responses are indicated in red below each respective comment. Please also note that line number references in the responses are for the track-change version enclosed herein.

GENERAL COMMENTS:

This manuscript presents an assessment study of the hydrologic and geologic impact on managed aquifer recharge processes. At 100 randomly sampled sites across the model domain the correlation between 17 hydro(geo)logical site characteristics/parameters and simulated recharge "benefits" is evaluated. Overall, upscaled vertical K multiplied with "Water Table Depth" (WTD) produce a good correlation with recharge rates. This proxy parameter (GPP – Kgeom * WTD) are most correlated with recharge rates, validated by local and global sensitivity analysis. Moreover, the analyses also indicate that permeability and unsaturated zone pore volume (porosity used as an indicator for Sy) were relatively more important than other hydraulic parameters.

The study presented is comprehensive, thorough, well-organized, and clear. The conclusions are informative and I do not see any over-statement in the conclusions drawn. I thus think the manuscript should be considered for publication in HESS, although I would suggest that a minor revision is needed to clarify some parts of the manuscript. Maybe the most critical point I see is how to transfer the obtain important information for MAR (interconnected coarse-texture facies paired with water table depth information are crucial for finding suitable recharge sites) to any other field where this information is difficult to acquire. I see your point that GIS-derived indices of recharge suitability rely solely on soil and surface geology to determine geologic suitability for recharge (e.g. Line 354) and the integrated values (up-scaled K + WTD) are more useful. But the question is how we could get the required information without knowing the subsurface in every detail in the whole model domain/study area. Certainly, in your (semi-)synthetic approach we know the parameterization (by the way it is just one field/realization and there remain uncertainties about the distribution, however, for this study and target it is ok I believe but should be note more clearly in the discussion) but how could we use your guidelines where it is not known. I suggest discussing that more to strengthen the manuscript and impact of this interesting study. Comprehensive field tests could be probably the best to better understand the problems in general. However, they are time-consuming and only a limited number of sites are available to accommodate the tests. The numerical analysis, on the other hand, allows us to explore and assess multiple sites relatively easily, yet the validity needs to be carefully checked. So, how can your useful guideline to be considered by practitioners? Another point is related to the WTD. Correct me if I am wrong but my impression based on your manuscript (for instance Fig. 8) is that Kgem * WTD is very useful where the WTD is deep (so large unsaturated zone and thus more storage). Can you split your analysis/results further to see if the depth to the water table matters or not?

RESPONSE: Thank you for the insightful comment. We agree that the manuscript would be strengthened by including discussion of potential approaches for characterizing the subsurface heterogeneity in real-world situations. We have added an additional paragraph in the Discussion to highlight some emerging geophysical approaches that show promise for characterizing subsurface geologic architecture for MAR (lines 497-506).

To the second point regarding splitting the analysis/results to determine whether the water table is important for recharge, the authors contend that we have discussed these findings in the manuscript (lines 372-374; Fig. 6), where we show that water-table depth alone is a poor predictor of recharge rate.

FURTHER MINOR COMMENTS:

COMMENT: Line 94: What kind of geological analysis?
RESPONSE: We agree that this language was ambiguous. We've have removed the first phrase of the sentence "Through geologic analysis of the data, additional parameters we estimated …" to just "Additional parameters were estimated" because the preceding text makes it clear that these parameters are part of the greater geostatistical analysis (line 94).

COMMENT: Section 2.3.3 Model Spin up and Calibration: All sentence related to boundary conditions should be moved to section 2.3.2 Boundary conditions. (line 134-141)
RESPONSE: We agree and have moved the five sentences describing boundary conditions to the preceding section (2.3.2) as a new paragraph (lines 131-135).

COMMENT: Table 2: Are these parameters the calibrated values?
RESPONSE: Yes, these are the calibrated values. The title of the table has been changed to "Calibrated Hydrofacies Hydraulic Properties" (Table 2).

COMMENT: Section 2.4.1 Why do you select the sites randomly? I would assume that MAR will be pretty much every time in more coarse sediments. I think, a useful comparison would be to choose the sites based on the surface information (as you mentioned as the "classical" GIS-approach) and compare the results with results from some randomly chosen sites. You might find additional arguments to criticize the "classical" workflow. I think that would be "just" another post-processing step and no demanding model runs are required.
RESPONSE: The intention of gathering 100 random sites from 910 potential sites was to represent the variability of geologic configuration throughout the domain in a computationally-efficient manner. It was not known a-priori which site characteristics would be best correlated with MAR, and our intention was not to presume the 'best' sites from within the domain. We contend that our approach clearly shows the limitations of the "classical workflow" of identifying sites with favorable surficial geology. Our results highlight the limitations of approaches that rely on surficial geology alone (lines 486-487 and lines 493-496). The approach suggested in the comment is explored in a companion paper recently published by the authors in Hydrogeology Journal (1)

1. Maples, S. R., Fogg, G. E., and Maxwell, R. M. (2019) Modeling Managed Aquifer Recharge Processes in a Highly Heterogeneous, Semi-Confined Aquifer System, Hydrogeology Journal, doi:10.1007/s10040-019-02033-9.

COMMENT: Line 201: for all 100 sites
RESPONSE: We agree that "all 100 recharge simulations" was confusing, and changed the text to "all 100 sites" (line 217).

COMMENT: Line 230: 6 hydraulic properties and not 8!
RESPONSE: Thank you for catching this mistake. "eight" has been changed to "six" accordingly (line 246).

COMMENT: Line 258: Where are the four representative sites! You could add these sites to figure 3a.
RESPONSE: We agree that highlighting these sites in Fig. 3a would be helpful and have modified the figure accordingly (line 173).

COMMENT: Line 286: yes, they are important, but it is not demonstrated here. The results section just comes a few pages later. Please reformulate.
RESPONSE: We agree that the results presented in the Methods section are out of place. We have re-phrased this paragraph to not include mention of results, and instead reframe the introduction of Morris parameters without mention of results (lines 300-307).

COMMENT: Line 357: Yes, but again how to get this information for a larger study site.
RESPONSE: We agree with the Reviewer's point, which is also stated in the General Comments, about the need for applicable field methods to make use of these findings. We have added paragraph of emerging geophysical techniques that could be used to validate findings presented here for real-world sites in the Discussion (lines 497-506).

COMMENT: Figure 7: Change the 95% confidence lines to dashed lines or change the figure caption.
RESPONSE: We contend that the dashed lines are clearly indicated as the 95% contour intervals in figure 7a, but we have added some clarifying language to the figure caption to make this more clear (line 383).

COMMENT: Figure 10: Why is the gravel n so important for V_fines?
RESPONSE: We agree that this is an interesting result, and it is only observed for site q95. We attribute this result to the fact that site q95 has a high proportion of gravel, so it is not unexpected that pore size distribution (n) of gravel would have some influence on the recharge response. We have made a change to the text to acknowledge this result (line 419), but do explore the implications in detail.

Notes to Reviewer 2: Author responses are indicated in red below each respective comment. Please also note that line number references in the responses are for the track-change version enclosed herein.

GENERAL COMMENTS

COMMENT: "The manuscript "Sensitivity of Hydrologic and Geologic Parameters on Recharge Processes in a Highly-Heterogeneous, Semi-Confined Aquifer System" describes an interesting study on local and global sensitivity analysis in the framework of Managed Aquifer recharge, using a realistic case study. Overall, the manuscript is well written and the results are illustrated in a clear manner. Although the research work heavily relies for the creation of the geological model and the setting up of a flow model and MAR on two previous works, the additional research performed in this study and the new findings justify a new publication. I only have a couple of minor suggestions and some technical details."
RESPONSE: Thank you for the comments. We have addressed each of your comments, and have provided responses to each specific comment below.

SPECIFIC COMMENTS:

COMMENT: "Control volume and connectivity metric (lines 179-188; 331-333) Please double check the definition of the control volume and the need for a 6-points connectivity metric: if the control volume is defined as "encompassing vertically-coincident cells" (line 179), then there is probably no need to require a 6-points connectivity metric. For example, with a 6-points connectivity, you can have 2 very horizontally extended layers of a conductive material, separated by a rather impermeable aquitard; if only one cell of the aquitard is conductive, then the 6-points connectivity guarantees connection. Maybe I missed the definition of the control volume. Is it defined by only one cell in the horizontal directions?"
RESPONSE: We agree that the definition of the control volume was unclear, and have made substantial changes to section 2.4.2 to re-frame and add detail to how site characteristics are presented. For example, we have added the sentence (lines 199-201): "Percolation was evaluated for a control volume encompassing all cells from the land surface to the initial water table depth (i.e., unsaturated-zone cells) at the 25 x,y cell locations encompassing each site." Because each control volume incorporates both vertically- and horizontally-connected cells, the 6-connectivity metric is necessary to evaluate percolation. We believe that the clarification regarding the definition of the control volume will make this clear to the reader.

COMMENT: "Linearity (197-200) As your aquifer is not confined, maybe the fact of separating the contribution of each recharge/no-recharge scenario would not work properly as in the case of a linear problem. Please comment on this."
RESPONSE: Thank you for pointing out some of the limitations associated with the differencing approach we use to post-process the results. We have added several sentences to the Discussion section to acknowledge the limitations of this approach for non-linear models, and also noted that we did not encounter spurious recharge stresses or unrealistic model noise when using this approach with our model. (lines 531-535)

COMMENT: "r sign In general, for a negative correlation a negative r is used (line 315, but also the corresponding figures)."
RESPONSE: Thank you for catching this mistake. We added text to point out that R_10d, R_30d, and P_30d were positively correlated with all simulated outputs, but V_fines, 90d was generally negatively correlated with simulated outputs (lines 354-57). We made changes throughout the manuscript to report negative correlations appropriately, and have modified Fig. 5 to make clear that correlations are reported as the absolute value (line 356).

FIGURES

COMMENT: "Fig.1 and Fig.3 Please report the original publication source of the figure."
RESPONSE: We have reported the original source of Figure 1 (Maples et al., 2019) (line 83), but Figure 3 is unique to this publication, and thus does not have a publication source, so the Figure 3 caption was left as-is (line 173).

COMMENT: "Fig.8 Do you also have a map of IVF? It would be nice to see it on the side of the R30d (see also line 370)."
RESPONSE: We chose not to overlay a map of IVF on our stochastic geologic model because our findings are presented as a proof-of-concept of a hypothetical, but physically-realistic domain (see lines 530-531, "Our findings are presented as a proof-of-concept to explore the importance of geologic heterogeneity on MAR in a hypothetical but physically-realistic domain"). Instead, we rely on citations of the relevant studies that have identified IVF in this region (e.g., lines 66-73) and encourage the reader to seek those publications for additional information.

TECHNICAL CORRECTIONS

COMMENT: "line 63 :1640 m2"
RESPONSE: Thank you for catching this mistake. We have changed "1640m^2" to "1640 m^2" (line 63).

COMMENT: "Parenthesis: Double check journal guidelines for parentheses (i.e. lines 67, 76-77, 87, 91)"
RESPONSE: We have checked the manuscript preparation guidelines for HESS and did not find specific guidance for these instances of parentheses. We will defer to the associate editor and copy editor to provide guidance on these instances.

COMMENT: "Units repetition. It would be more correct to report units close to each number, for example $1 \times 2 \times 3$ m should be something like $1$ m $\times 2$ m $\times 3$ m (see lines 99, 101). This is also valid when a list of numbers (with unit) is reported. See for example line 308, 309, 320."
RESPONSE: Thank you for catching this mistake. We have made changes here and elsewhere throughout the manuscript to correct unit repetition issues (lines 99, 101).

COMMENT: "Subscript fonts In general, subscripts that are not index should not be in italic font (i.e., $S_s$ should be $S_s$ instead) (see line 110 and other locations in the text)"

RESPONSE: Thank you for catching this mistake. We have made changes here and elsewhere throughout the manuscript to correct italicization mistakes in subscripts (lines 110, 112-117, and elsewhere).

COMMENT: "lines 123-124 Check '0 m amsl'."
RESPONSE: We are leaving the acronym as-is because above mean sea level (amsl) is introduced previously in Section 2.1 (line 77).

COMMENT: "UZ (line 141) Please introduce this acronym."
RESPONSE: Thank you for catching this mistake. We have changed "near-surface UZ cells" to "near-surface unsaturated-zone (UZ) cells" (lines 142-143).

COMMENT: "line 165-166 $S_s$ or $SS$?"
RESPONSE: Thank you for catching this mistake. We have changed "$K\_S$" to "$K\_s$" (line 173).

COMMENT: "$V_{fines,90d}$ Double check the consistency of this symbol within the documents (see for example Fig.5)."
RESPONSE: Thank you for catching this mistake. We have fixed the figure accordingly (line 356), and have double-checked the consistency of its usage throughout the text.

line 479 "to be fully..."?
RESPONSE: Thank you for catching this mistake. We have changed "challenging to fully captured" with "challenging to fully capture" (line 509).

line 538 "to incorporate a measure"
RESPONSE: Thank you for catching this mistake. We have incorporated the edit (line 572).

[revised manuscript text omitted]
 as well as the 4 sites chosen for sensitivity analyses (q95, q75, q50, and q25; solid violet squares), along with (b) arithmetic, (c) geometric, and (d) harmonic mean of vertical $K_{sat}$ for unsaturated zone (UZ) facies (i.e., $K_{geom}$, $K_{arith}$, $K_{harm}$, respectively). The second row shows (e) the coarse-texture (gravel and sand) fraction of UZ facies ($UZ_{coarse}$), (f) simulated initial depth-to-water ($WTD$), and (g) $K_{geom}$ multiplied by $WTD$ ($K_{geom} \times WTD$). Locations >5 km from the lateral domain boundaries were excluded from the potential sites to avoid interference with boundary conditions. *(double-column width)*

41,087 x,y cell locations across the domain, where the arithmetic and harmonic mean ($K_{arith}$ and $K_{harm}$) are the upper and lower bounds, respectively, and the geometric mean ($K_{geom}$) is an intermediate value. $K_{arith}$ and $K_{harm}$ are typically used to approximate groundwater flow parallel and perpendicular to layering, respectively, in anisotropic systems (Freeze and Cherry, 1979). This concept has been generally been extended to variably-saturated flow  (Mualem, 1984; Yeh et al., 1985a, b; Assouline and Or, 2006). Fogg et al. (2000) showed that vertical groundwater flow in

systems with vertically-connected networks of permeable facies tends toward values between $K_{\text{arith}}$ and $K_{\text{geom}}$.  For

180 each x,y cell location across the domain, $K_{\text{arith}}$, $K_{\text{geom}}$, and $K_{\text{harm}}$  are given as:

$$K_{\text{arith}} = \frac{K_1 + K_2 + ... + K_{\text{n}}}{n} \tag{3}$$

$$K_{\text{geom}} = \sqrt[n]{K_1 \times K_2 \times ... \times K_{\text{n}}} \tag{4}$$

$$K_{\text{harm}} = \frac{n}{\frac{1}{K_1} + \frac{1}{K_2} + ... + \frac{1}{K_{\text{n}}}} \tag{5}$$

185  where $n$ is the number of vertically-coincident cells from the land surface to the initial water table depth (i.e., the unsaturated-zone cells) for each x,y cell location. In addition, the initial unsaturated-zone thickness ($WTD$)  and proportion of coarse-texture sand and gravel unsaturated-zone facies ($UZ_{\text{coarse}}$)

190  at each x,y cell location across the domain were included as metrics. Additional metrics were developed for each x,y cell location by combining individual metrics, i.e., $WTD$ was used as a multiplier for $K_{\text{arith}}$, $K_{\text{geom}}$, $K_{\text{harm}}$, $Surf_{\text{coarse}}$, and $UZ_{\text{coarse}}$. Spatial distributions of select  metrics at all x,y cell locations are shown in Fig. 3.

*TABLE 3 ABOUT HERE.*

195 Site characteristics were calculated for the 100 1 km$^2$ exploratory sites from these metrics by determining the average and maximum value of $K_{\text{arith}}$, $K_{\text{geom}}$, and $K_{\text{harm}}$ for the 25 x,y cell locations encompassing each site. In addition, the 
[revised manuscript text omitted]

[Figure]

**Figure 4.** Box plots of the (a) 30-day average recharge rate, $R_{30d}$ and (b) 30-day pressure perturbation area of influence, $R_{30d}$ for all exploratory simulations ($n = 100$). Additionally, sites were parsed according to whether there was vertical interconnection of coarse-texture facies from land surface to the initial water table depth (i.e., interconnected sites, $n = 23$), or whether sites did not have interconnection of coarse-texture facies (i.e., disconnected sites, $n = 77$).

[Figure]

**Figure 5.** Correlations ([absolute value of]{.underline} Pearson's $r$) for all combinations of site characteristics and model outputs. Correlations among site characteristics are bounded by a solid red box, and correlations between site characteristics and model outputs are bounded by a dashed red box.

[simulated outputs.]{.underline} Strong correlation ( $|r| > 0.70$) was observed for 6 of 52 pairs of site characteristics and simulated outputs. Strong correlation was also observed among many site characteristics and among the majority of simulated outputs (i.e, collinearity), which can make the choice of an optimal proxy parameter more challenging. Site characteristics that include 360 $K_{\text{harm}}$ were not shown in the correlation matrix because we were not able to improve normality of the distribution these data with a $\text{Log}_{10}$ data transformation; however, additional correlation metrics (Fig. 6) indicate that site characteristics that include $K_{\text{harm}}$ may also be strongly correlated.

**3.2.2 Ranked Correlations**

Additional correlation metrics (Pearson's $r$, Spearman's rho, and Kendall's tau) between $R_{30d}$ and site characteristics were 365 ranked and are shown in Fig. 6. Results show that site characteristics that include some variation of $K_{\text{arith}}$, $K_{\text{geom}}$, or $K_{\text{harm}}$ were, in general, more correlated with $R_{30d}$ than site characteristics that only include $WTD$, $UZ_{\text{coarse}}$, and $Surf_{\text{coarse}}$. $K_{\text{geom}} \times WTD$ was, on average, most correlated with $R_{30d}$.

In general, site characteristics that included $K_{\text{geom}}$ and $K_{\text{harm}}$ were slightly more correlated with $R_{30d}$ than site characteristics that included $K_{\text{arith}}$. We speculate that this behavior is related to the dominantly vertical flow direction of recharge across

[revised manuscript text omitted]

‡ Baseline hydraulic properties were calibrated manually by Liu (2014)